# Exploring Non-Contrastive Representation Learning for Deep Clustering

## Abstract

Existing deep clustering methods rely on contrastive learning for representation learning, which requires negative examples to form an embedding space where all instances are well-separated. However, the negative examples inevitably give rise to the class collision issue, compromising the representation learning for clustering. In this paper, we explore non-contrastive representation learning for deep clustering, termed NCC, which is based on BYOL, a representative method *without* negative examples. First, we propose to align one augmented view of instance with the neighbors of another view in the embedding space, called positive sampling strategy, which avoids the class collision issue caused by the negative examples and hence improves the within-cluster compactness. Second, we propose to encourage alignment between two augmented views of one prototype and uniformity among all prototypes, named prototypical contrastive loss or ProtoCL, which can maximize the inter-cluster distance. Moreover, we formulate NCC in an Expectation-Maximization (EM) framework, in which E-step utilizes spherical k-means to estimate the pseudo-labels of instances and distribution of prototypes from a target network and M-step leverages the proposed losses to optimize an online network. As a result, NCC forms an embedding space where all clusters are well-separated and within-cluster examples are compact. Experimental results on several clustering benchmark datasets including ImageNet-1K demonstrate that NCC outperforms the state-of-the-art methods by a significant margin.

## 1 Introduction

Deep clustering is gaining considerable attention as it can learn representation of images and perform clustering in an end-to-end fashion. Remarkably, contrastive learning-based methods (Wang et al., 2021; Van Gansbeke et al., 2020; Li et al., 2021a;b; Tao et al., 2021; Tsai et al., 2021; Niu et al., 2021) have become the main thrust to advance the representation of images on several complex benchmark datasets, significantly contributing to the clustering performance. In addition, some contrastive learning methods such as MoCo (He et al., 2020) and SimCLR (Chen et al., 2020) usually require specially designed losses (Wang et al., 2021; Li et al., 2021a;b; Tao et al., 2021; Tsai et al., 2021) or an extra pre-training stage for more discriminative representations (Van Gansbeke et al., 2020; Niu et al., 2021).

Although achieving promising clustering results, contrastive learning requires a large number of negative examples to achieve the instance-wise discrimination in an embedding space where all instances are well-separated. The constructed negative pairs usually require a large batch size (Chen et al., 2020), memory queue (He et al., 2020), or memory bank (Wu et al., 2018), which not only bring extra computational cost but also give rise to class collision issue (Saunshi et al., 2019). Here, the class collision issue refers to that different instances from the same semantic class are regarded as negative pairs, hurting the representation learning for clustering. A question naturally arises: *are negative examples necessary for deep clustering?*

Another kind of self-supervised learning is the non-contrastive methods such as BYOL (Grill et al., 2020) and SimSiam (Chen & He, 2021), which use the representations of one augmented view to predict another view. Their success demonstrates that negative examples are not the key to avoiding representation collapse. However, to the best of our knowledge, almost all recent successful literature of deep clustering is built upon contrastive learning-based methods such as MoCo (He et al.,

2020) and SimCLR (Chen et al., 2020). There is a general consensus that the negative examples are helpful to stabilize the training of representation learning for deep clustering. As discussed in (Wang & Isola, 2020), the typical contrastive loss can be identified into two properties: 1) *alignment* term to improve the closeness of positive pairs; and 2) *uniformity* term to encourage instances to be uniformly distributed on a unit hypersphere. In contrast, non-contrastive methods such as BYOL *only* optimize the alignment term, leading to unstable training and suffering from the representation collapse—which may be worsen when adding extra losses.

To tackle the class collision issue, we explore the non-contrastive representation learning for deep clustering, termed non-contrastive clustering or NCC, which is based on BYOL, a representative method *without* negative examples. First, instead of negative sampling that is a double-edged sword, *i.e.*, causing class collision issue but improving the training stability, we propose to align one augmented view of the instance with the neighbors of another view in the embedding space, called positive sampling strategy, which can avoid the class collision issue and hence improve the within-cluster compactness. Second, as for the clustering task, the different clusters are truly negative pairs for contrastive loss. To this end, we propose to encourage the alignment between two augmented views of prototypes and the uniformity among all prototypes, named prototypical contrastive loss or ProtoCL, which can maximize the inter-cluster distance. Moreover, we formulate our method into an EM framework, in which we iteratively perform E-step as estimating the pseudo-labels of instances and distribution of prototypes via spherical k-means based on the target network and M-step as optimizing the online network via the proposed losses. As a result, NCC is able to form an embedding space where all clusters are well-separated and within-cluster examples are compact. The contributions of this paper are summarized as follows:

- We explore the non-contrastive representation learning for deep clustering by proposing non-contrastive clustering or NCC, which is based on the Bootstrap Your Own Latent (BYOL), a representative method *without* negative examples.

- We propose a positive sampling strategy to augment instance alignment by taking into account neighboring positive examples in the embedding space, which can avoid the class collision issue and hence improve the within-cluster compactness.

- We propose a novel prototypical contrastive loss or ProtoCL, which can align one augmented view of prototypes with another view and encourage the uniformity among all prototypes on a unit hypersphere, hence maximizing the inter-cluster distance.

- We formulate our method into an EM framework, in which we can iteratively estimate the pseudo-labels and distribution of prototypes via spherical k-means based on the target network and optimize the online network via the proposed losses.

- Extensively experimental results on several benchmark datasets as well as ImageNet-1K demonstrate that NCC outperforms the existing state-of-the-art methods by a significant margin.

## 2   RELATED WORK

Deep clustering can be significantly advanced by discriminative representations. Examples of traditional deep clustering methods include: Xie et al. (2016); Yang et al. (2017) use autoencoders to simultaneously perform representation learning and clustering; Chang et al. (2017); Haeusser et al. (2018); Wu et al. (2019); Ji et al. (2019) learn pair-wise relationships between original and augmented instances. However, they often suffer from inferior performance on some complex datasets such as CIFAR-20. Inspired by the success of contrastive learning, recent studies turn to exploit the discriminative representations learned from contrastive learning to assist the downstream clustering tasks (Van Gansbeke et al., 2020; Niu et al., 2021) or simultaneously optimize representation learning and clustering (Tao et al., 2021; Tsai et al., 2021; Li et al., 2021a; Shen et al., 2021). SCAN (Van Gansbeke et al., 2020) uses the model pre-trained by SimCLR to yield the confident pseudo-labels. IDFD (Tao et al., 2021) proposes to perform both instance discrimination and feature decorrelation. GCC (Zhong et al., 2021) and WCL (Zheng et al., 2021) build a graph to label the neighbor samples as pseudo-positive examples, however, they still suffer from the class collision issue due to the contrastive loss involved and these pseudo-positive examples that may not be truly positive. All of them are built upon the contrastive learning framework, which means that they require a large number of negative examples for training stability, inevitably giving rise to class

collision issue. Different from prior work, this paper explores the non-contrastive self-supervised methods, *i.e.*, BYOL, to achieve both representation learning and clustering. We note that Regatti et al. (2021); Lee et al. (2020) have tried to build the clustering framework based on BYOL, however, their methods do not consider improving within-cluster compactness and maximizing inter-cluster distance like ours. Therefore, to the best of our knowledge, this is the first successful attempt that introduces the non-contrastive representation learning into deep clustering that yields a substantial performance improvement over previous state-of-the-art methods. In Appendix A, we present related work on self-supervised learning and difference from existing methods including CC (Li et al., 2021b), GCC (Zhong et al., 2021), WCL (Zheng et al., 2021), and PCL (Li et al., 2021a).

## 3 PRELIMINARY

The most successful self-supervised learning methods in recent years can be roughly divided into contrastive (Chen et al., 2020; He et al., 2020) and non-contrastive (Grill et al., 2020; Chen & He, 2021). Here, we briefly summarize their formulas and discuss their difference.

**Contrastive learning.** Contrastive learning methods perform instance-wise discrimination (Wu et al., 2018) using the InfoNCE loss (Oord et al., 2018). Formally, assume that we have one instance $x$, its augmented version $x^+$ by using random data augmentation, and a set of $M$ negative examples drawn from the dataset , $\{x_1^-, x_2^-, \ldots, x_M^-\}$. The contrastive learning aims to learn an embedding function $f$ that maps $x$ onto a unit hypersphere, in which the InfoNCE loss can be defined as:

$$\mathcal{L}_{\text{contr}} = -\log \frac{\exp\left(f(x)^{\text{T}} f(x^+)/\tau\right)}{\exp\left(f(x)^{\text{T}} f(x^+)/\tau\right) + \sum_{i=1}^{M} \exp\left(f(x)^{\text{T}} f(x_i^-)/\tau\right)} \tag{1}$$

$$\approx -f(x)^{\text{T}} f(x^+)/\tau + \log \sum_{i=1}^{M} \exp\left(f(x)^{\text{T}} f(x_i^-)/\tau\right), \tag{2}$$

where the first and second terms in Eq. (2) refer to as instance alignment and instance uniformity, respectively. Here, we assume that the output of $f(\cdot)$ is $\ell_2$ normalized. That is, the representation is on a unit hypersphere. The temperature $\tau$ controls the concentration level of representations; please refer to (Wang & Liu, 2021) for detailed behaviors of $\tau$ in the contrastive loss. Intuitively, the InfoNCE loss aims to pull together the positive pair $(x, x^+)$ from two different data augmentations of the same instance, and push $x$ away from $M$ negative examples of other instances. As discussed in (Wang & Isola, 2020), when $M \to \infty$, the InfoNCE loss in Eq. (1) can be approximately decoupled into two terms: *alignment* and *uniformity*, as shown in Eq. (2). Despite the alignment term closes the positive pair, the key to avoiding representation collapse is the uniformity term, which makes the negative examples uniformly distributed on the hypersphere. Although beneficial, the negative examples inevitably lead to the class collision issue, hurting the representation learning for clustering.

**Non-contrastive learning.** Non-contrastive learning-based methods have shown more promising results than contrastive learning for representation learning and downstream tasks (Ericsson et al., 2021). Non-contrastive methods only optimize the alignment term in Eq. (2) to match the representations between two augmented views. Without negative examples, they leverage an online and a target network for two views, and use a predictor network to bridge the gap between these two views. They also stop the gradient of the target network to avoid the representation collapse. In particular, if $\tau = 0.5$, the loss used in (Grill et al., 2020; Chen & He, 2021) can be written as:

$$\mathcal{L}_{\text{non}-\text{contr}} = -2g\left(f(x)\right)^{\text{T}} f'(x^+) = \left\|g\left(f(x)\right) - f'(x^+)\right\|_2^2 + \text{const}, \tag{3}$$

where $g$ the predictor; $f$ and $f'$ are the online and target networks, respectively; the outputs of $g(f(\cdot))$ and $f'(\cdot)$ are $\ell_2$-normalized. However, as mentioned in (Fetterman & Albrecht, 2020), the non-contrastive learning methods often suffer from unstable training and highly rely on the batch-statistics and hyper-parameter tuning to avoid representation collapse. Even though Grill et al. (2020); Richemond et al. (2020) have proposed to use some tricks such as SyncBN (Ioffe & Szegedy, 2015) and weight normalization (Qiao et al., 2019) to alleviate this issue, the additional computation cost is significant. Without negative examples, the collapse issue could be worsen when adding additional clustering losses for clustering task; see Fig. A1 for the analysis of applying PCL (Li et al., 2021a) to the BYOL.

In a nutshell, most of existing successful deep clustering methods are based on contrastive learning for representation learning—giving rise to class collision issue—while the non-contrastive learning, due to unstable training with additional losses, is not yet ready for deep clustering. To that end, we explore the non-contrastive learning, *i.e.* BYOL, for deep clustering with positive sampling strategy and prototypical contrastive loss to avoid the class collision issue, improve the within-cluster compactness, and maximize the inter-class distance.

# 4 NON-CONTRASTIVE CLUSTERING

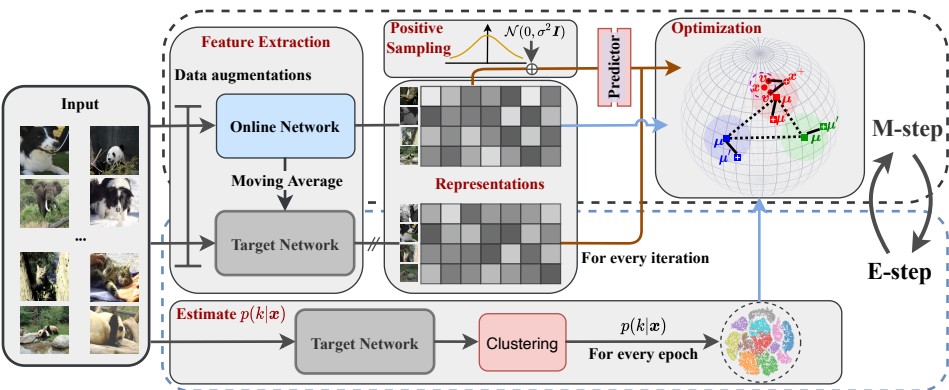

Figure 1: The overall framework of the proposed NCC in an EM framework.

Fig. 1 presents the overall framework of the proposed NCC. Based on BYOL, NCC is comprised of three networks: an online, a target, and a predictor. In Sec. 4.1, we propose a positive sampling strategy to augment instance alignment to improve the within-cluster compactness. In Sec. 4.2, a prototypical contrastive loss is introduced to maximize the inter-cluster distance using the pseudo-labels from k-means clustering, which can encourage uniform representations. Finally, we formulate NCC into an EM framework to facilitate the understanding of training procedure in Sec. 4.3.

## 4.1 POSITIVE SAMPLING STRATEGY

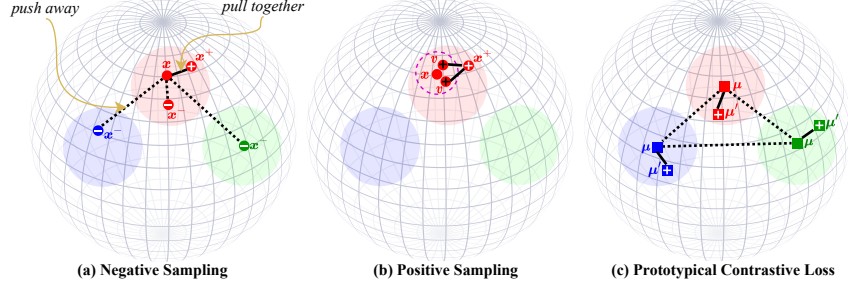

(a) Negative Sampling     (b) Positive Sampling     (c) Prototypical Contrastive Loss

Figure 2: Illustration of the proposed techniques compared to negative sampling. (a) Negative sampling in contrastive learning giving rise to class collision issue. (b) The proposed positive sampling encouraging the alignment between neighbors of one view with another one. (c) The proposed prototypical contrastive loss encouraging prototypical alignment and prototypical uniformity.

The negative examples are essential for contrastive learning-based deep clustering to stabilize the training of representation learning, at the cost of inevitable class collision issue (Saunshi et al., 2019), as shown in Fig. 2(a). This issue can hurt the representation learning for clustering as the instances from the same class/cluster—should be close to each other—could be treated as negative pairs and are pushed away during training, discouraging the within-cluster compactness.

To address the class collision issue, we resort to non-contrastive learning-based methods for representation learning, which no longer need negative examples. Although we cannot optimize the

uniformity term like contrastive loss, our idea is to optimize the *opposite* of the uniformity instead. That is, we aim to encourage the neighboring examples around one augmented view, sampled in the embedding space, to be aligned with another view, as shown in Fig. 2(b). Our motivation is that although we cannot guarantee the negative examples in contrastive loss are from different classes, we can *certainly* assume that the neighboring samples around one view in the embedding space are positive with respect to another view and belong to the same class. Therefore, we propose a positive sampling strategy to augment the instance alignment in Eq. (3) by taking into account the neighboring samples towards within-cluster compactness.

Specifically, we model the representation of one augmented view of an instance as a Gaussian distribution in the embedding space, which can be formulated as follows:

$$\boldsymbol{v} \sim \mathcal{N}\left(f(\boldsymbol{x}), \sigma^2 \boldsymbol{I}\right), \tag{4}$$

where $\boldsymbol{I}$ represents the identity matrix and $\sigma$ is a positive hyperparameter controlling how many samples around one view can be treated as positive pairs with another view. However, the sampled examples from Eq. (4) cannot allow the error to be backpropagated through the network to update the network parameters. We employ the reparametrization trick (Kingma & Welling, 2014) to achieve the backpropagation. As a result, the positive sampling strategy can be implemented as follows:

$$\boldsymbol{v} = f(\boldsymbol{x}) + \sigma\boldsymbol{\epsilon}, \quad \boldsymbol{\epsilon} \sim \mathcal{N}\left(0, \boldsymbol{I}\right). \tag{5}$$

Therefore, we can augment the instance alignment in Eq. (3) by taking into account the neighboring samples to encourage the within-cluster compactness. With only sampling one example from the Gaussian distribution, the augmented instance alignment term can be formally defined as:

$$\mathcal{L}_{\text{aug-ins}} = \left\| g(\boldsymbol{v}) - f'(\boldsymbol{x}^+) \right\|_2^2 = \left\| g\left(f(\boldsymbol{x}) + \sigma\boldsymbol{\epsilon}\right) - f'(\boldsymbol{x}^+) \right\|_2^2. \tag{6}$$

The benefits of the proposed positive sampling are summarized as follows.

**Improved within-cluster compactness.** The conventional instance alignment in Eq. (3) only encourages the representation of one augmented view to be close to another view. In the context of clustering, such compactness is instance-wise and neutral for the clustering. Put differently, all instances are treated as cluster centers and the semantic class information cannot be captured at only instance level. In contrast, our augmented instance alignment in Eq. (6) encourages neighboring examples around one augmented view—either different augmented examples of the same instance or same/different augmented examples of different instances within the same cluster—to be positive pairs with another view. This is helpful to improve within-cluster compactness.

**Avoidable class collision issue.** As we mentioned before, class collision issue induced by the negative examples indicates that we cannot guarantee that the negative examples are from different clusters. However, our positive sampling strategy can guarantee that the positive examples around one instance, sampled in the embedding space, are from the same cluster as the instance, getting rid of class collision issue. The difference from recent work (Zhong et al., 2021; Zheng et al., 2021) is discussed in Appendix A. We note that our positive sampling strategy does not consider uniformity, which is solved by the proposed prototypical contrastive loss in Sec. 4.2.

## 4.2 PROTOTYPICAL CONTRASTIVE LOSS

A good clustering is supposed to have distinct semantic prototypes/clusters. Assume that the dataset has $K$ clusters, where $K$ is a predefined hyperparameter, it naturally constructs a contrastive loss for these $K$ prototypes as for one prototype, the remaining $K-1$ prototypes are definitely negative examples. Therefore, we propose a prototypical contrastive loss or ProtoCL, which encourages the prototypical alignment between two augmented views and the prototypical uniformity, hence maximizing the inter-cluster distance.

Specifically, assume we have $K$ prototypes from the online network, $\{\boldsymbol{\mu}_1, \boldsymbol{\mu}_2, \dots, \boldsymbol{\mu}_K\}$, and another $K$ prototypes from the target network, $\{\boldsymbol{\mu}'_1, \boldsymbol{\mu}'_2, \dots, \boldsymbol{\mu}'_K\}$, our proposed ProtoCL, illustrated in Fig. 2(c), is given as follows:

$$\mathcal{L}_{\text{pcl}} = \frac{1}{K}\sum_{k=1}^{K} -\log \frac{\exp(\boldsymbol{\mu}_k^{\mathrm{T}}\boldsymbol{\mu}'_k/\tau)}{\exp(\boldsymbol{\mu}_k^{\mathrm{T}}\boldsymbol{\mu}'_k/\tau) + \sum_{j=1, j\neq k}^{K}\exp(\boldsymbol{\mu}_k^{\mathrm{T}}\boldsymbol{\mu}_j/\tau)}, \tag{7}$$

$$\approx \frac{1}{K}\sum_{k=1}^{K} -\boldsymbol{\mu}_k^{\mathrm{T}}\boldsymbol{\mu}'_k/\tau + \frac{1}{K}\sum_{k=1}^{K}\log\sum_{j=1, j\neq k}^{K}\exp(\boldsymbol{\mu}_k^{\mathrm{T}}\boldsymbol{\mu}_j/\tau), \tag{8}$$

where the first and second terms in Eq. (8) refer to as prototypical alignment and prototypical uniformity, respectively. Here, the cluster centers $\boldsymbol{\mu}_k$ and $\boldsymbol{\mu}'_k$ are computed within a mini-batch $\mathcal{B}$ as follows:$\boldsymbol{\mu}_k = \frac{\sum_{\boldsymbol{x} \in \mathcal{B}} p(k|\boldsymbol{x}) f(\boldsymbol{x})}{\| \sum_{\boldsymbol{x} \in \mathcal{B}} p(k|\boldsymbol{x}) f(\boldsymbol{x}) \|_2}$ and $\boldsymbol{\mu}'_k = \frac{\sum_{\boldsymbol{x} \in \mathcal{B}} p(k|\boldsymbol{x}) f'(\boldsymbol{x})}{\| \sum_{\boldsymbol{x} \in \mathcal{B}} p(k|\boldsymbol{x}) f'(\boldsymbol{x}) \|_2}$, and $p(k|\boldsymbol{x})$ is the cluster assignment posterior probability. When $K > |\mathcal{B}|$, it is obvious that the mini-batch cannot cover all clusters. To this end, we zero out the losses and logits of empty clusters for each iteration; see the pseudocode in Appendix D for more details.

Clearly, our ProtoCL is quite similar to conventional contrastive loss in Eq. (1) but for prototypes with non-contrastive representation learning framework. The prototypical alignment is to align the prototypes derived from the online network with the ones from the target network, which can stabilize the update of the prototypes. The prototypical uniformity is to encourage the prototypes to be uniformly distributed on a unit hypersphere, which can maximize the inter-cluster distance. The difference from ProtoNCE in (Li et al., 2021a) is discussed in Appendix A.

### 4.3 EM FRAMEWORK

We formulate our NCC into an EM framework to facilitate the understanding of the training procedure, detailed in Fig. 1 and derived in Appendix B.

**E-step.** This step aims to estimate $p(k|\boldsymbol{x})$. We perform spherical k-means algorithm on the features extracted from the target network since the target network performs more stable and yields more consistent clusters, similar to BYOL and MoCo. Although we need an additional k-means clustering to obtain the cluster pseudo-labels $p(k|\boldsymbol{x})$ for every $r$ epochs, we found that even with a larger $r$, rather than every epoch $r = 1$, our method can still produce consistent performance improvement over the baseline methods. Therefore, our method will not introduce much computation cost and is robust to the cluster pseudo-labels; see detailed results in Fig. A2. The analysis of computational cost is discussed in Appendix C. Finally, with $p(k|\boldsymbol{x})$, we build the cluster centers within a mini-batch without additional memory like queue (He et al., 2020) or bank (Wu et al., 2018).

**M-step.** Combining the augmented instance alignment loss in Eq. (6) and the proposed ProtoCL in Eq. (7) yields our objective function for M-step as follows:

$$\mathcal{L} = \mathcal{L}_{\mathrm{aug-ins}} + \lambda_{\mathrm{pcl}} \mathcal{L}_{\mathrm{pcl}}, \tag{9}$$

where $\lambda_{\mathrm{pcl}}$ controls the balance between two loss components. Therefore, there are only two additional hyper-parameters compared to original BYOL, including: $\sigma$ in $\mathcal{L}_{\mathrm{aug-ins}}$ and the loss weight $\lambda_{\mathrm{pcl}}$; see detailed results of these two hyper-parameters in Figs. A3 and A4.

## 5 EXPERIMENTS

We conducted experiments on six benchmark datasets, including **CIFAR-10** (Krizhevsky et al., 2009), **CIFAR-20** (Krizhevsky et al., 2009), **STL-10** (Coates et al., 2011), **ImageNet-10** (Chang et al., 2017), **ImageNet-Dogs** (Chang et al., 2017), and **ImageNet-**

Table 1: Summary of the datasets.

| Dataset | Split | # Samples | # Classes | Image Size |
|---|---|---|---|---|
| CIFAR-10 | Train+Test | 60,000 | 10 | 32×32 |
| CIFAR-20 | Train+Test | 60,000 | 20 | 32×32 |
| STL-10 | Train+Test | 13,000 | 10 | 96×96 |
| ImageNet-10 | Train | 13,000 | 10 | 96×96 |
| ImageNet-Dogs | Train | 19,500 | 15 | 96×96 |
| ImageNet-1K | Train | 1,281,167 | 1,000 | 224×224 |

**1K** (Deng et al., 2009), which are summarized in Table 1. We note that CIFAR-20 contains 20 superclasses of CIFAR-100. This paper follows the experimental settings widely used in deep clustering work (Chang et al., 2017; Wu et al., 2019; Ji et al., 2019; Tsai et al., 2021; Tao et al., 2021), including the image size, backbone and train-test split. We employ three common metrics to evaluate the clustering performance, including Normalized Mutual Information (NMI), Cluster Accuracy (ACC), and Adjusted Rand Index (ARI) for the first five datasets. Following (Li et al., 2021a), we report Adjusted Mutual Information (AMI) to evaluate the clustering performance for ImageNet-1K. The results are presented in percentage (%) and the higher the better clustering performance. For fair comparisons, we use ResNet-34 (He et al., 2016) as the backbone to report the results in Table 2. Unless noted otherwise, we use ResNet-18 for the rest of experiments. We run each experiment three times and report the mean and standard deviation as the final results. We provided detailed training settings in Appendix C. We also provide the pseudocode of NCC for better understanding in Appendix D. The source code will be publicly available upon acceptance.

## 5.1 MAIN RESULTS

Table 2: Clustering results (%) of various methods on five benchmark datasets. The best and second best results are shown in bold and underline, respectively.

| Dataset | CIFAR-10 | | | CIFAR-20 | | | STL-10 | | | ImageNet-10 | | | ImageNet-Dogs | | |
|---|---|---|---|---|---|---|---|---|---|---|---|---|---|---|---|
| Method[1] | NMI | ACC | ARI | NMI | ACC | ARI | NMI | ACC | ARI | NMI | ACC | ARI | NMI | ACC | ARI |
| k-means | 8.7 | 22.9 | 4.9 | 8.4 | 13.0 | 2.8 | 12.5 | 19.2 | 6.1 | 11.9 | 24.1 | 5.7 | 5.5 | 10.5 | 2.0 |
| SC | 10.3 | 24.7 | 8.5 | 9.0 | 13.6 | 2.2 | 9.8 | 15.9 | 4.8 | 15.1 | 27.4 | 7.6 | 3.8 | 11.1 | 1.3 |
| AE | 23.9 | 31.4 | 16.9 | 10.0 | 16.5 | 4.8 | 25.0 | 30.3 | 16.1 | 21.0 | 31.7 | 15.2 | 10.4 | 18.5 | 7.3 |
| VAE | 24.5 | 29.1 | 16.7 | 10.8 | 15.2 | 4.0 | 20.0 | 28.2 | 14.6 | 19.3 | 33.4 | 16.8 | 10.7 | 17.9 | 7.9 |
| JULE | 19.2 | 27.2 | 13.8 | 10.3 | 13.7 | 3.3 | 18.2 | 27.7 | 16.4 | 17.5 | 30.0 | 13.8 | 5.4 | 13.8 | 2.8 |
| DEC | 25.7 | 30.1 | 16.1 | 13.6 | 18.5 | 5.0 | 27.6 | 35.9 | 18.6 | 28.2 | 38.1 | 20.3 | 12.2 | 19.5 | 7.9 |
| DAC | 39.6 | 52.2 | 30.6 | 18.5 | 23.8 | 8.8 | 36.6 | 47.0 | 25.7 | 39.4 | 52.7 | 30.2 | 21.9 | 27.5 | 11.1 |
| IIC | 51.3 | 61.7 | 41.1 | - | 25.7 | - | 43.1 | 49.9 | 29.5 | - | - | - | - | - | - |
| DCCM | 49.6 | 62.3 | 40.8 | 28.5 | 32.7 | 17.3 | 37.6 | 48.2 | 26.2 | 60.8 | 71.0 | 55.5 | 32.1 | 38.3 | 18.2 |
| PICA | 56.1 | 64.5 | 46.7 | 29.6 | 32.2 | 15.9 | - | - | - | 78.2 | 85.0 | 73.3 | 33.6 | 32.4 | 17.9 |
| CC[2] | 70.5 | 79.0 | 63.7 | 43.1 | 42.9 | 26.6 | 76.4 | 85.0 | 72.6 | 85.9 | 89.3 | 82.2 | 44.5 | 42.9 | 27.4 |
| SCAN[3] | 79.7 | 88.3 | 77.2 | 48.6 | 50.7 | 33.3 | 80.9 | 69.8 | 64.6 | - | - | - | - | - | - |
| GCC | 76.4 | 85.6 | 72.8 | 47.2 | 47.2 | 30.5 | 68.4 | 78.8 | 63.1 | 84.2 | 90.1 | 82.2 | 49.0 | 52.6 | 36.2 |
| MiCE | 73.7 | 83.5 | 69.8 | 43.6 | 44.0 | 28.0 | 63.5 | 75.2 | 57.5 | - | - | - | 42.3 | 43.9 | 28.6 |
| IDFD | 71.1 | 81.5 | 66.3 | 42.6 | 42.5 | 26.4 | 64.3 | 75.6 | 57.5 | **89.8** | 95.4 | 90.1 | 54.6 | 59.1 | 41.3 |
| PCL | 80.2 | 87.4 | 76.6 | 52.8 | 52.6 | 36.3 | 71.8 | 41.0 | 67.0 | 84.1 | 90.7 | 82.2 | 44.0 | 41.2 | 29.9 |
| BYOL | 81.7 | 89.4 | 79.0 | 55.9 | 56.9 | 39.3 | 71.3 | 82.5 | 65.7 | 86.6 | 93.9 | 87.2 | 63.5 | 69.4 | 54.8 |
| | ±0.1 | ±0.6 | ±0.1 | ±0.3 | ±1.8 | ±0.2 | ±0.9 | ±0.5 | ±1.3 | ±0.2 | ±0.1 | ±0.2 | ±2.2 | ±3.0 | ±2.9 |
| NCC | **88.6** | **94.3** | **88.4** | **60.6** | **61.4** | **45.1** | **75.8** | **86.7** | **73.7** | 89.6 | **95.6** | **90.6** | **69.2** | **74.5** | **62.7** |
| (ours) | ±0.1 | ±0.6 | ±1.1 | ±0.3 | ±1.1 | ±0.1 | ±1.8 | ±1.3 | ±2.4 | ±0.2 | ±0.0 | ±0.1 | ±0.3 | ±0.1 | ±0.1 |

[1] k-means (Lloyd, 1982), SC (Zelnik-manor & Perona, 2005), AE (Bengio et al., 2007), VAE (Kingma & Welling, 2014),
JULE (Yang et al., 2016), DEC (Xie et al., 2016), DAC (Chang et al., 2017), IIC (Ji et al., 2019), DCCM (Wu et al., 2019),
PICA (Huang et al., 2020), CC (Li et al., 2021b), SCAN (Van Gansbeke et al., 2020), GCC (Zhong et al., 2021),
MiCE (Tsai et al., 2021), IDFD (Tao et al., 2021), PCL (Li et al., 2021a), BYOL (Grill et al., 2020),
[2] CC uses a large image size (224) for all datasets.
[3] SCAN needs an additional pre-training stage while NCC is trained in an end to end manner. It only uses training set for all datasets.

**Quantitative results** We compared NCC with previous state-of-the-art clustering methods in Table 2. NCC achieves significant performance improvement on all five benchmark datasets, demonstrating the superiority of NCC for deep clustering to capture the semantic class information. Interestingly, directly using the representations learned by BYOL for k-means clustering outperforms previous work including the contrastive-based ones (Li et al., 2021b; Tsai et al., 2021; Tao et al., 2021), which suggests a great potential for non-contrastive representation learning for deep clustering *without* suffering class collision issue.

On the ImageNet-10, our NCC achieves competitive performance as compared to IDFD (Tao et al., 2021) since this dataset is relatively small with only 13K images, which cannot arise discriminative differences for current state-of-the-art methods. On the ImageNet-Dogs, a fine-grained dataset containing different species of dogs from the ImageNet dataset, there are almost 20% improvements over previous SOTA work. The contrastive-based methods cannot handle this kind of dataset due to severe class collision issue that pushes away the instances from the same class. Meanwhile, IDFD can deal with this problem to some degree thanks to the feature decorrelation along with the instance discrimination. Without the need of negative examples, BYOL can achieve significant improvement, although its performance is unstable. Our NCC, built upon BYOL with a positive sampling strategy and prototypical contrastive loss, has shown its significant and stable performance against vanilla BYOL and contrastive-based methods.

Table 3 further presents the results between our NCC and baseline methods including DeepCluster (Caron et al., 2018), MoCo (He et al., 2020), and PCL (Li et al., 2021a) on ImageNet-1K dataset, showing that NCC achieves significantly higher AMI score.

Although we employed the fair conditions, some work has trained the network with different split (Van Gansbeke et al., 2020) for CIFAR-10 and CIFAR-20, or large image size (Li et al., 2021b) for ImageNet-10 and ImageNet-Dogs. For the sake of fair comparisons with different image sizes and splits, we report these comparison results in Table A1. In addition, we also reported the clustering results on ImageNet subsets like SCAN (Van Gansbeke et al., 2020)

Table 3: Clustering results (%) on ImageNet-1K.

| Method | AMI |
|---|---|
| DeepCluster | 28.1 |
| MoCo | 28.5 |
| PCL | 41.0 |
| NCC (Ours) | **52.5** |

 We also conducted additional experiments in Table A4 to demonstrate the ability of NCC handling the long-tailed datasets.

**Qualitative results.** Fig. 3 visualizes the learned representations by t-SNE (Van der Maaten & Hinton, 2008) for four different training epochs throughout the training process. At the beginning, the random-initialized model cannot distinguish the instances from different semantic classes, where all instances are mixed together. As the training process goes, NCC gradually attracts the instances from the same cluster while pushing the clusters away from each other. Obviously, at the end of the training, NCC produces clear boundary between clusters and within-cluster compactness. Visualization for the outlier points produced by the model at 1000-th epoch is shown in Fig. A5.

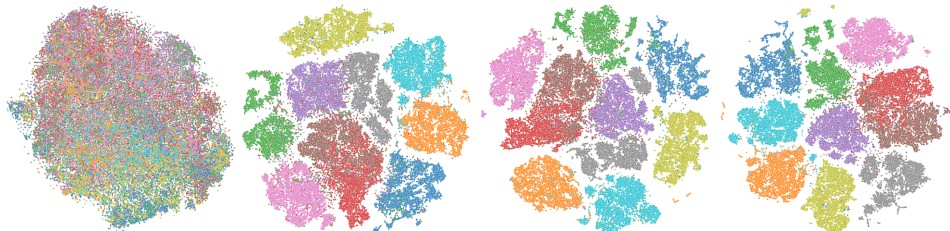

a) Epoch 0 (NMI=6.98%)  b) Epoch 300 (NMI=77.8%)  c) Epoch 700 (NMI=83.5%)  d) Epoch 1000 (NMI=85.5%)

Figure 3: Visualization of feature representations learned by NCC on CIFAR-10 with t-SNE. Different colors denote the different semantic classes. Zoom in for better view.

## 5.2 ABLATION STUDY

Here, we perform detailed ablation studies with both quantitative and qualitative comparisons to provide more insights into why NCC performs so well for deep clustering.

Table 4: Ablation studies for different self-supervised learning frameworks, and positive sampling (PS) strategy, and prototypical contrastive loss for NCC. The best and second best results are shown in bold and underline, respectively.

| Method | PS | Prototypical | | CIFAR-10 | | | CIFAR-20 | | |
| | | Alignment | Uniformity | NMI | ACC | ARI | NMI | ACC | ARI |
| --- | --- | --- | --- | --- | --- | --- | --- | --- | --- |
| MoCo v2 (He et al., 2020) | | | | 76.9±0.2 | 84.9±0.3 | 72.4±0.5 | 49.2±0.1 | 48.0±0.2 | 32.1±0.0 |
| SimSiam (Chen & He, 2021) | | | | 78.8±0.9 | 86.5±0.8 | 74.9±1.3 | 46.6±0.8 | 47.3±1.1 | 28.8±1.2 |
| CC (Li et al., 2021b) | | | | 66.1±0.3 | 74.6±0.3 | 58.3±0.4 | 46.4±0.3 | 45.0±0.1 | 29.5±0.2 |
| + ProtoCL (Ours) | | ✓ | ✓ | 74.3±0.4 | 83.4±0.5 | 69.6±1.0 | 48.3±0.2 | 49.1±0.2 | 32.2±0.4 |
| PCL (Li et al., 2021a) | | | | 77.6±0.1 | 85.5±0.1 | 73.4±0.0 | 50.0±0.3 | 48.6±0.7 | 32.7±0.4 |
| BYOL (Grill et al., 2020) | | | | 79.4±1.7 | 87.8±1.7 | 76.6±2.8 | 55.5±0.6 | 53.9±1.6 | 37.6±0.9 |
| + CC (Li et al., 2021b) | | | | 76.6±3.1 | 86.3±2.7 | 73.8±4.7 | 51.0±2.0 | 48.9±3.0 | 33.3±2.9 |
| + PCL (Li et al., 2021a) | | | | 74.4±2.3 | 85.3±0.9 | 71.4±1.4 | 49.7±0.7 | 46.9±0.7 | 27.8±1.5 |
| | ✓ | | | 79.4±0.9 | 87.9±0.5 | 76.4±1.1 | 57.0±0.0 | 55.0±0.6 | 39.8±1.1 |
| | | ✓ | ✓ | 83.4±1.2 | 90.3±0.9 | 81.1±1.7 | 56.6±0.4 | 55.1±0.5 | 40.7±1.0 |
| NCC (Ours) | ✓ | ✓ | | 79.6±0.7 | 87.8±1.5 | 76.5±2.1 | 56.7±0.3 | 56.6±1.4 | 39.7±1.1 |
| | ✓ | | ✓ | **85.3**±0.2 | **92.1**±0.1 | **84.4**±0.3 | 57.2±0.3 | 57.3±0.6 | 41.7±0.5 |
| | ✓ | ✓ | ✓ | 85.1±0.5 | 91.6±0.4 | 83.5±0.7 | **58.2**±0.3 | **57.8**±0.2 | **42.3**±0.3 |

**Quantitative ablation study.** We report the quantitative results of ablation studies in Table 4. BYOL outperforms MoCo v2 and SimSiam by a large margin on both two datasets. The difference between BYOL and MoCo v2 is that MoCo v2 uses a memory queue to store the consistent negative examples while BYOL directly aligns two augmented views with a predictor network. Different from the BYOL that employs a momentum-updated network as the target network to yield the positive representations, SimSiam shares the weights of the target and online networks. Therefore, BYOL outperforms MoCo v2 by dealing with the class collision issue and SimSiam by the momentum-updated target network.

Compared to vanilla BYOL, simply using the positive sampling strategy can stable and further improve the performance, especially when the number of semantic classes increases for CIFAR-20. Although ProtoCL improves the baseline results by a large margin, positive sampling can further boost the clustering performance. This is because ProtoCL only considers inter-cluster distance, and cannot benefit within-cluster compactness. Therefore, the combination of the positive sampling

and ProtoCL achieves the best clustering results, where positive sampling strategy can improve the within-cluster compactness and ProtoCL encourages the prototypical alignment between two augmented views and maximizes the inter-cluster distance.

To further explore the effect of the proposed ProtoCL, we split this loss function into prototypical alignment and uniformity as shown in Eq. (8). It is clear that the performance gain from the alignment term is marginal while the gain from the uniformity term is significant. Note that for only alignment term, we compute the loss after predictor network instead of feature extractor, otherwise representation collapse will turn out. This indicates that prototypical uniformity is more important than prototypical alignment since BYOL has already performed instance alignment at an augmented instance level. However, we note that the prototypical alignment term is essential to stabilize the training process, as demonstrated in the results for CIFAR-20 with more clusters.

To demonstrate that BYOL is not robust to additional losses for deep clustering tasks, we integrate CC (Li et al., 2021b) and PCL (Li et al., 2021a) into BYOL; the results in Table 4 show that both of them compromise clustering performance and become unstable. This is because CC contrasts the cluster probability not helpful for representation learning, and the representations of PCL would collapse without negative examples; see Fig. A1 for detailed analysis. For fair comparisons of the self-supervised learning framework, we integrate our ProtoCL into CC by replacing the cluster head with our ProtoCL on the representations while keeping other official hyper-parameters unchanged. Although class collision issue remains, the significant improvements over CC on both datasets suggest that class-level contrastive loss over representations of cluster centers is better than the one over cluster probabilities and ProtoCL can be generalized to other self-supervised learning frameworks.

**Qualitative ablation study.** Fig. 4 visualizes the distribution of representations learned from MoCo v2, BYOL, NCC w/o ProtoCL, and NCC. The representations from MoCo v2 are mixed up at the center due to the class collision issue. The rest can reduce this phenomenon where NCC w/o ProtoCL produces more compact clusters than BYOL, and NCC further maintains distinct borders between different clusters. In addition, we also visualize the training and clustering results of BYOL and NCC in Fig. A7, demonstrating that NCC can produce more uniform representations, more balanced clusters, and better clustering performance than BYOL.

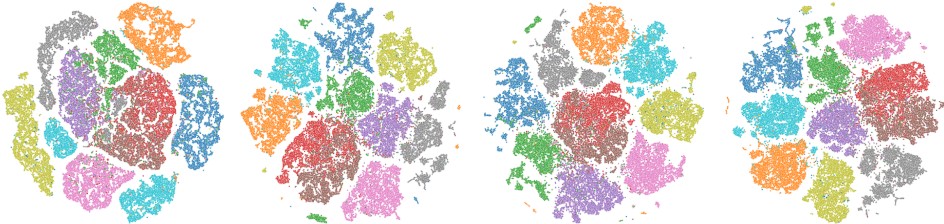

a) MoCo (NMI=76.6%)      b) BYOL (NMI=78.4%)      c) NCC w/o ProtoCL (NMI=78.8%)      d) NCC (NMI=85.5%)

Figure 4: Visualization of feature representations learned by different representation learning frameworks and our proposed NCC on CIFAR-10 with t-SNE. Zoom in for better view.

**Additional ablation studies.** To explore the influences of different hyper-parameters in NCC, we perform the following ablation studies: 1) performing k-means clustering for every $r$ epochs in Fig. A2; 2) $\sigma$ in positive sampling in Fig. A3; 3) $\lambda_{\mathrm{pcl}}$ for ProtoCL in Fig. A4; 4) predefined number of clusters $K$ in Fig. A6; 5) projection dimension for self-supervised learning in Fig. A8; 6) data augmentation in Fig. A9 for self-supervised learning; and 7) different ResNet architectures in Table A5. All ablation study results verify that the performance gain of NCC does not come from backbone, projection dimension, or any other hyper-parameters. The results also suggest that NCC is robust to the choice of hyper-parameters.

## 6  CONCLUSION

We have explored the non-contrastive representation learning for deep clustering. The proposed positive sampling strategy and prototypical contrastive loss can lead to within-cluster compactness and well-separated clusters towards the goodness of clustering. The results suggest that the proposed NCC outperforms the state-of-the-art methods by a significant margin. We hope our study will attract the community's attention to the non-contrastive representation learning methods for deep clustering, which do not suffer from class collision issue.

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

APPENDICES

# A  ADDITIONAL RELATED WORK

**Self-supervised learning.**  Previous self-supervised learning (SSL) methods for representation learning attempt to capture the data distribution using generative models (Donahue et al., 2017; Donahue & Simonyan, 2019) or learn the representations through some special designed pretext tasks (Doersch et al., 2015; Noroozi & Favaro, 2016; Zhang et al., 2016; Caron et al., 2018). In recent years, contrastive learning methods (Wu et al., 2018; He et al., 2020; Chen et al., 2020) have shown promising results for both representation learning and downstream tasks. For example, MoCo (He et al., 2020) uses a memory queue to store the consistent representations output by a moving-averaged encoder. However, the class collision issue remains unavoidable; *i.e.*, the semantic similar instances are pushed away since they could be regarded as negative pairs (Saunshi et al., 2019). Some attempts have been made to address this issue (Khosla et al., 2020; Hu et al., 2021; Chuang et al., 2020). On the contrary, the recent studies of SSL demonstrate that the negative examples are not necessary, termed non-contrastive methods (Caron et al., 2020; Grill et al., 2020; Chen et al., 2020). In summary, SSL methods mainly focus on inducing transferable representations for the downstream tasks instead of grouping the data into different semantic classes in deep clustering.

**Relation to CC.**  Although both $\mathcal{L}_{\mathrm{pcl}}$ and CC (Li et al., 2021b) are class-level contrastive loss, which perform contrastive learning at the cluster level, they have the following difference.

- The class-level contrastive loss in CC implements the contrastive loss on the cluster probabilities while ours on the representation of cluster centers. Implementing contrastive loss on the cluster probability in (Li et al., 2021b) would lose the semantic information of the learned representations, which is not helpful for representation learning. Specifically, given the $x \in \mathcal{B}$, CC obtains the cluster assignments $\boldsymbol{P}_k = [p(k|\boldsymbol{x}^{(1)}), \ldots, p(k|\boldsymbol{x}^{(N)})]$ from one view and $\boldsymbol{P}_k'$ from another view, and then contrasts $\boldsymbol{P}_k$ and $\boldsymbol{P}_k'$ at the cluster level using the InfoNCE loss. In contrast, $\mathcal{L}_{\mathrm{pcl}}$ implements the contrastive loss on the representation of the cluster centers within a mini-batch using the pseudo-labels from k-means clustering. As a result, $\mathcal{L}_{\mathrm{pcl}}$ is able to sense the semantic information of the latent space and make the representations of clusters more discriminative and suitable for the clustering task.

- The class-level contrastive loss in CC does not encourage cluster uniformity while our ProtoCL does. CC still needs the instance-wise contrastive loss to encourage the instance uniformity, which inevitably introduces the class collision issue. We have also conducted experiments that integrate CC into BYOL and reported the results in Table 4. The performance of BYOL drops and the training becomes unstable. Under the same conditions, NCC achieves significant improvements over CC.

**Relation to GCC and WCL.**  GCC (Zhong et al., 2021) and WCL (Zheng et al., 2021) built a graph to label the neighbor samples as pseudo-positive examples. Then, they enforce the two data augmentations of one example to be close to its multiple pseudo-positive examples using a supervised contrastive loss. GCC adopted a moving-averaged memory bank for the graph-based pseudo-labeling while WCL built the graph within a mini-batch. GCC and WCL mainly focus on how to effectively select positive examples from mini-batch/memory bank to alleviate the class collision issue. Here, we divide the class collision issue into the following two cases:

- Negative class collision issue: negative examples may not be truly negative, which is the case we discussed in the paper.

- Positive class collision issue: positive examples may not be truly positive, which is a new case raised in GCC and WCL.

Consequently, they still suffer from the positive class collision issue as the selected pseudo-positive examples may not be truly positive. In addition to this, they also suffer from the negative class collision issue since they still need negative examples for instance-wise contrastive learning.

We summarize the difference between our positive sampling strategy and theirs in the following four aspects.

- GCC and WCL select the examples that exist in the dataset (mini-batch/memory bank) while ours samples augmented examples from the latent space that may not exist in the dataset.
- GCC and WCL select neighbor examples in a graph as pseudo-positive examples that may not be truly positive while ours samples examples around the instance that can be assumed to be positive in the semantic space.
- GCC and WCL still rely on instance-wise contrastive loss that could lead to class collision issue while ours can avoid class collision issue by using BYOL.
- GCC and WCL require additional computational cost for graph construction while ours is rather cheap in sampling one augmented example.

**Relation to PCL.**   Here, we summarize the difference between our NCC and PCL (Li et al., 2021a) in terms of the losses and EM frameworks. First, we summarize the difference of our ProtoCL loss and ProtoNCE loss in PCL as follow.

- Our NCC can avoid class collision issue while PCL cannot. NCC is based on BYOL that does not require negative examples for representation learning while PCL is based on instance-wise contrastive loss that requires a number of negative examples for representation learning, inevitably leading to class collision issue.
- The proposed ProtoCL in NCC is conceptually different from the ProtoNCE in PCL. ProtoCL is to maximize the inter-cluster distance to form a uniformly distributed space while ProtoNCE is to minimize the instance-to-cluster distance to improve the within-cluster compactness. The within-cluster compactness of NCC is improved by the proposed positive sampling strategy.
- Pure ProtoCL can work well for deep clustering while ProtoNCE requires another InfoNCE to form uniformly distributed space. This is a direct result of the different designs of the losses. ProtoCL can maximize the inter-cluster distance to form a uniformly distributed space while ProtoNCE suffers from collapse without the help of another InfoNCE to form such a space.

Second, we summarize difference between our NCC and PCL in the EM framework. Formulating NCC into an EM framework can offer more insights about NCC and make it easy to understand. Although both in an EM framework, the M-step in PCL is significantly different from the one in our NCC. More specifically, the M-step in PCL is to optimize the ProtoNCE, which is an **instance-to-prototypes contrastive loss** to improve the within-cluster compactness while the M-step in our NCC is to optimize the proposed ProtoCL, which is a **prototypes-to-prototypes contrastive loss** to maximize the inter-cluster distance for better clustering performance. In addition, NCC also proposes a positive sampling strategy by sampling positive examples around each sample to improve within-cluster compactness. Finally, Table 3 demonstrates that NCC outperforms PCL by almost 10% AMI on the ImageNet-1K dataset.

## B   EM FRAMEWORK

In this section, we first describe the *von Mises-Fisher* (vMF) distribution on the hypersphere, and then derive the ELBO for our EM framework, followed by detailed E-step and M-step. Finally, we describe our proposed prototypical contrastive loss and provide proof for convergence analysis.

**Von Mises-Fisher distribution.**   Since the features in current SSL methods are usually $\ell_2$-normalized, it is more proper to employ the spherical distribution to describe the features. The *von Mises-Fisher* (vMF) distribution, often seen as the Gaussian distribution on a hypersphere, is parameterized by $\boldsymbol{\mu} \in \mathbb{R}^d$ the mean direction and $\kappa \in \mathbb{R}_+$ the concentration around $\boldsymbol{\mu}$. For the special case of $\kappa = 0$, the vMF distribution represents a uniform distribution on the hypersphere. The PDF of vMF distribution for the random unit vector $\boldsymbol{v} \in \mathbb{R}^d$ is defined as:

$$p(\boldsymbol{v} \mid \boldsymbol{\mu}, \kappa) = \mathcal{C}_d(\kappa) \exp(\kappa \boldsymbol{\mu}^{\mathrm{T}} \boldsymbol{v}); \qquad \mathcal{C}_d(\kappa) = \frac{\kappa^{d/2-1}}{(2\pi)^{d/2} \mathcal{I}_{d/2-1}(\kappa)}, \tag{A1}$$

where $d$ is the feature dimension, $\|\boldsymbol{\mu}\|^2 = 1$, $\mathcal{C}_d(\kappa)$ is the normalizing constant, and $\mathcal{I}_v$ denotes the modified Bessel function of the first kind at order $v$. The standard Gaussian distribution $\boldsymbol{z} \sim \mathcal{N}(0, \boldsymbol{I})$ can be approximately seen as the uniform vMF distribution if the $\boldsymbol{z}$ is $\ell_2$-normalized and $d$ is large for the high dimension data.

**Derivation of ELBO.** Given the dataset $\mathcal{D} = \{\boldsymbol{x}^{(n)}\}_{n=1}^N$ with $N$ observed data points that are related to a set of $K$ cluster latent variables, $k \in \mathcal{K} = \{1, 2, \ldots, K\}$, the marginal likelihood can be written as:

$$\mathcal{L}(\mathcal{D}; \boldsymbol{\theta}) = \frac{1}{N} \sum_{n=1}^N \log p(\boldsymbol{x}^{(n)}; \boldsymbol{\theta}) = \frac{1}{N} \sum_{n=1}^N \log \sum_{k \in \mathcal{K}} p(\boldsymbol{x}^{(n)}, k; \boldsymbol{\theta}), \tag{A2}$$

where $\boldsymbol{\theta}$ denotes the model parameters. Eq. (A2) is usually maximized to train the neural network. However, it is hard to directly optimize the log-likelihood function. Using an inference model $q(k)$ like VAE (Kingma & Welling, 2014) to approximate the distribution of $\mathcal{K}$, especially $\sum_{k \in \mathcal{K}} q(k) = 1$, we can re-write the log-likelihood function for one example as:

$$\log p(\boldsymbol{x}; \boldsymbol{\theta}) = \sum_{k \in \mathcal{K}} q(k) \log p(\boldsymbol{x}; \boldsymbol{\theta}) \tag{A3}$$

$$= \sum_{k \in \mathcal{K}} q(k)(\log p(\boldsymbol{x}, k; \boldsymbol{\theta}) - \log p(k|\boldsymbol{x}; \boldsymbol{\theta})) \tag{A4}$$

$$= \sum_{k \in \mathcal{K}} q(k) \log \frac{p(\boldsymbol{x}, k; \boldsymbol{\theta})}{q(k)} - \sum_{k \in \mathcal{K}} q(k) \log \frac{p(k|\boldsymbol{x}; \boldsymbol{\theta})}{q(k)} \tag{A5}$$

$$= \text{ELBO}(q, \boldsymbol{x}; \boldsymbol{\theta}) + \text{KL}(q(k)\|p(k|\boldsymbol{x}; \boldsymbol{\theta})), \tag{A6}$$

where $p(\boldsymbol{x}, k; \boldsymbol{\theta}) = p(k|\boldsymbol{x}; \boldsymbol{\theta})p(\boldsymbol{x}; \boldsymbol{\theta})$ so we have $\log p(\boldsymbol{x}; \boldsymbol{\theta}) = \log p(\boldsymbol{x}, k; \boldsymbol{\theta}) - \log p(k|\boldsymbol{x}; \boldsymbol{\theta})$ and the evidence lower bound (ELBO) is the lower bound of log-likelihood function since $\text{KL}(q(k)\|p(k|\boldsymbol{x}; \boldsymbol{\theta})) \geq 0$. When $\text{KL}(q(k)\|p(k|\boldsymbol{x}; \boldsymbol{\theta})) = 0$, the ELBO reaches its maximum value $\log p(\boldsymbol{x}; \boldsymbol{\theta})$, making $q(k) = p(k|\boldsymbol{x}; \boldsymbol{\theta})$. By replacing $q(k)$ with $p(k|\boldsymbol{x}; \boldsymbol{\theta})$ and ignoring the constant value $\sum_{k \in \mathcal{K}} -q(k) \log q(k)$, we are ready to maximize:

$$\sum_{k \in \mathcal{K}} p(k|\boldsymbol{x}; \boldsymbol{\theta}) \log p(\boldsymbol{x}, k; \boldsymbol{\theta}). \tag{A7}$$

**E-step.** With the fixed $\boldsymbol{\theta}_t$ at the iteration $t$, this step aims to estimate $q_{t+1}(k)$ that makes $q_{t+1}(k) = p(k|\boldsymbol{x}; \boldsymbol{\theta}_t)$ so that $\text{ELBO}(q_{t+1}, \boldsymbol{x}; \boldsymbol{\theta}_t) = \log p(\boldsymbol{x}; \boldsymbol{\theta}_t)$. Here, we perform the spherical k-means algorithm to estimate $p(k|\boldsymbol{x}; \boldsymbol{\theta}_t)$. We extract features from the target network since the target network performs more stable and yields more consistent clusters, similar to BYOL and MoCo.

**M-step.** With the fixed suboptimal $q_{t+1}(k) = p(k|\boldsymbol{x}; \boldsymbol{\theta}_t)$ after E-step, we turn to optimize the $\boldsymbol{\theta}$ to maximize the ELBO:

$$\boldsymbol{\theta}_{t+1} = \arg\max_{\boldsymbol{\theta}} \sum_{n=1}^N \text{ELBO}(q_{t+1}, \boldsymbol{x}^{(n)}; \boldsymbol{\theta}). \tag{A8}$$

Using a uniform prior for $k$ as $p(k) = 1/K$, we can obtain $p(\boldsymbol{x}, k; \boldsymbol{\theta}) = p(k)p(\boldsymbol{x}|k; \boldsymbol{\theta}) = p(\boldsymbol{x}|k; \boldsymbol{\theta})/K$. By replacing $p(\boldsymbol{x}, k; \boldsymbol{\theta})$ in Eq. (A7) and ignoring constant value, in this step, we should maximize:

$$\sum_{k \in \mathcal{K}} \mathbb{1}(\boldsymbol{x} \in k) \log p(\boldsymbol{x}|k; \boldsymbol{\theta}), \tag{A9}$$

where $p(k|\boldsymbol{x}; \boldsymbol{\theta}) = \mathbb{1}(\boldsymbol{x} \in k)$. $\mathbb{1}(\cdot)$ is an indicator function using the hard labels estimated from E-step so that $\mathbb{1}(\boldsymbol{x} \in k) = 1$ if $\boldsymbol{x}$ belongs to $k$-th cluster; otherwise $\mathbb{1}(\boldsymbol{x} \in k) = 0$. Following (Li et al., 2021a), if we assume that the distribution for each cluster is the vMF distribution with a constant $\kappa$ as the temperature of softmax function, we can further obtain the follow:

$$p(\boldsymbol{x}|k; \boldsymbol{\theta}) = \frac{\exp(\boldsymbol{\mu}_k^{\mathrm{T}} \boldsymbol{v}/\tau)}{\sum_{k=1}^K \exp(\boldsymbol{\mu}_k^{\mathrm{T}} \boldsymbol{v}/\tau)}, \tag{A10}$$

where $\boldsymbol{v} = f(\boldsymbol{x}; \boldsymbol{\theta})$, $\tau = 1/\kappa$, and $\boldsymbol{\mu}_k$ is the cluster center of $k$-th cluster. Combining Eqs. (A9) and (A10), we can achieve the maximum log-likelihood estimation to find the optimal $\boldsymbol{\theta}^*$ by minimizing the following negative log-likelihood:

$$\boldsymbol{\theta}^* = \arg\min_{\boldsymbol{\theta}} \sum_{n=1}^N -\log \frac{\exp(\boldsymbol{\mu}_{y^{(n)}}^{\mathrm{T}} \boldsymbol{v}^{(n)}/\tau)}{\sum_{k=1}^K \exp(\boldsymbol{\mu}_k^{\mathrm{T}} \boldsymbol{v}^{(n)}/\tau)}, \tag{A11}$$

where $y^{(n)}$ is the pseudo-label for $\boldsymbol{x}^{(n)}$ estimated by the k-means algorithm in E-step.

Directly optimizing Eq. (A11) usually leads to improve the cluster compactness, which, however, will compromise the stability of BYOL since it does not consider the uniformity term. To this end, we propose a prototypical contrastive loss (ProtoCL) to maximize the log-likelihood at the cluster level by employing the clusters centers as the special instances, or prototypes from a set of instances. The ProtoCL is defined as:

$$\mathcal{L}_{\text{pcl}} = \frac{1}{K}\sum\nolimits_{k=1}^{K} -\log \frac{\exp(\boldsymbol{\mu}_k^{\mathrm{T}}\boldsymbol{\mu}_k'/\tau)}{\exp(\boldsymbol{\mu}_k^{\mathrm{T}}\boldsymbol{\mu}_k'/\tau) + \sum_{j=1,j\neq k}^{K}\exp(\boldsymbol{\mu}_k^{\mathrm{T}}\boldsymbol{\mu}_j/\tau)}, \tag{A12}$$

where $\{\boldsymbol{\mu}_1, \boldsymbol{\mu}_2, \ldots, \boldsymbol{\mu}_K\}$ and $\{\boldsymbol{\mu}_1', \boldsymbol{\mu}_2', \ldots, \boldsymbol{\mu}_K'\}$ are $K$ prototypes from target and online networks, respectively. Here, instead of using the centroids computed from k-means, our cluster center $\boldsymbol{\mu}_k$ and $\boldsymbol{\mu}_k'$ is empirically estimated within mini-batch $\mathcal{B}$ as follows:

$$\boldsymbol{\mu}_k = \frac{\sum_{\boldsymbol{x}\in\mathcal{B}} p(k|\boldsymbol{x})f(\boldsymbol{x})}{\|\sum_{\boldsymbol{x}\in\mathcal{B}} p(k|\boldsymbol{x})f(\boldsymbol{x})\|_2} \quad \text{and} \quad \boldsymbol{\mu}_k' = \frac{\sum_{\boldsymbol{x}\in\mathcal{B}} p(k|\boldsymbol{x})f'(\boldsymbol{x})}{\|\sum_{\boldsymbol{x}\in\mathcal{B}} p(k|\boldsymbol{x})f'(\boldsymbol{x})\|_2}, \tag{A13}$$

where $p(k|\boldsymbol{x})$ is estimated from E-step. When $K > |\mathcal{B}|$, it is obvious that the mini-batch cannot cover all clusters. To this end, we zero out the losses and logits of empty clusters for each iteration; see the pseudocode in Appendix D for more details.

Intuitively, ProtoCL can encourage the prototypical alignment between two augmented views and the prototypical uniformity, hence maximizing the inter-cluster distance.

**Convergence analysis.** At E-step of the iteration $t$, we estimate $q_{t+1}(k)$ to make $\text{ELBO}(q_{t+1}, \boldsymbol{x}; \boldsymbol{\theta}_t) = \log p(\boldsymbol{x}; \boldsymbol{\theta}_t)$. At M-step after E-step, we obtain the optimized $\boldsymbol{\theta}_{t+1}$ with the fixed $q_{t+1}(k)$ so that $\text{ELBO}(q_{t+1}, \boldsymbol{x}; \boldsymbol{\theta}_{t+1}) \geq \text{ELBO}(q_{t+1}, \boldsymbol{x}; \boldsymbol{\theta}_t)$. Consequently, we obtain the following sequence:

$$\log p(\boldsymbol{x}; \boldsymbol{\theta}_{t+1}) \geq \text{ELBO}(q_{t+1}, \boldsymbol{x}; \boldsymbol{\theta}_{t+1}) \geq \text{ELBO}(q_{t+1}, \boldsymbol{x}; \boldsymbol{\theta}_t) = \log p(\boldsymbol{x}; \boldsymbol{\theta}_t). \tag{A14}$$

Given $\log p(\boldsymbol{x}; \boldsymbol{\theta}_{t+1}) \geq \log p(\boldsymbol{x}; \boldsymbol{\theta}_t)$, we can guarantee the convergence of our NCC.

## C  EXPERIMENTAL SETUP

**Datasets.** We conducted experiments on six benchmark datasets, including **CIFAR-10** (Krizhevsky et al., 2009), **CIFAR-20** (Krizhevsky et al., 2009), **STL-10** (Coates et al., 2011), **ImageNet-10** (Chang et al., 2017), **ImageNet-Dogs** (Chang et al., 2017), **ImageNet-1K** (Deng et al., 2009), which are summarized in Table 1. We note that CIFAR-20 contains 20 superclasses of CIFAR-100. STL-10 includes extra unlabeled images. ImageNet-10 and ImageNet-Dogs are the subset of ImageNet-1K, containing 10 and 15 classes, respectively. This paper follows the experimental settings widely used in deep clustering work (Chang et al., 2017; Wu et al., 2019; Ji et al., 2019; Tsai et al., 2021; Tao et al., 2021), including the image size, backbone and train-test split. For image size, we have used $32 \times 32$ for CIFAR-10 and CIFAR-20, $96 \times 96$ for STL-10, ImageNet-10 and ImageNet-Dogs, and $224 \times 224$ for ImageNet-1K. For train-test split, we use the whole datasets including training and testing set for CIFAR-10 and CIFAR-20.

**Backbone.** We use ResNet-34 (He et al., 2016) as the backbone for fair comparisons to report the results in Table 2. Unless noted otherwise, we use ResNet-18 for the rest of the experiments. Since the image sizes of CIFAR-10 and CIFAR-100 are relatively small, following (Chen et al., 2020), we replace the first convolution layer of kernel size $7 \times 7$ and stride 2 with a $3 \times 3$ Conv of stride 1 and remove the first max-pooling layer for all experiments on CIFAR-10 and CIFAR-100.

**Metrics.** We employ three common metrics to evaluate the clustering performance for the former five datasets, including Normalized Mutual Information (NMI), Cluster Accuracy (ACC), and Adjusted Rand Index (ARI). Following (Li et al., 2021a), we report Adjusted Mutual Information (AMI) to evaluate the clustering performance for ImageNet-1K. All the metrics are presented in percentage (%), and the higher the better clustering performance. We run each experiment three times and report the mean and standard deviation as the final results.

**Optimization.** We train all methods with 1000 epochs, strictly following the literature (Tsai et al., 2021; Tao et al., 2021), and adopt the SGD optimizer and the cosine decay learning rate schedule

with 50 epochs for learning rate warmup. The base learning rate for MoCo v2, BYOL, and NCC were 0.05, scaled linearly with the batch size (LearningRate = 0.05×BatchSize/256). Note that the learning rates for predictor networks of BYOL and NCC are $10\times$ as the learning rate of feature extractor. It is relatively important to achieve satisfactory performance, as discussed in (Grill et al., 2020; Chen & He, 2021). For other hyperparameters of NCC, the temperature $\tau$, $\lambda_{\mathrm{pcl}}$ for prototypical contrastive loss, and $\sigma$ for positive sampling were set as 0.5, 0.1, and 0.001, respectively. The mini-batch size was 512 for MoCo and 256 for the rest methods, trained on 4 NVIDIA V100 GPUs. In terms of the results of CC (Li et al., 2021b) and PCL (Li et al., 2021a) in Table 4, we tried our best to reproduce their results for fair comparisons. For CC, we used their official code. For PCL, under the fair conditions of MoCo, we set the loss weight of ProtoNCE to 0.01 and the number of clusters to $\{250, 500, 1000\}$ following the suggestions of authors, which we found can achieve the best results. We integrated CC and PCL into BYOL by adding their losses without changing other settings.

**Configurations of self-supervised learning frameworks.** We adopt the same data augmentations as SimCLR (Chen et al., 2020), including ResizedCrop, ColorJitter, Grayscale, and HorizontalFlip. We have removed GaussianBlur since we only used a small image size for all datasets. We also strictly follow the settings of BYOL (Grill et al., 2020). Specifically, despite the standard ResNet backbones, the projection and predictor networks have the architectures of FC-BN-ReLU-FC, where the projection dimension and hidden size were 256 and 4096 for both two networks, respectively. In the context of the whole paper, we have included the backbone and projection network as the encoder network, or feature extractor, since the architectures except predictor are symmetric across views. For fair comparisons, we have also set the projection dimension of MoCo v2 as 256. We have used symmetric loss for all methods, *i.e.*, swapping two data augmentations to compute twice loss. Given the target network $f_{\boldsymbol{\theta}'}'$ and online network $f_{\boldsymbol{\theta}}$, the momentum rule describes that the $\boldsymbol{\theta}'$ is updated as $\boldsymbol{\theta}' = m\boldsymbol{\theta}' + (1-m)\boldsymbol{\theta}$ with a momentum hyperparameter $m \in [0, 1)$. We set the $m$ to 0.996 for both BYOL and NCC same as (Grill et al., 2020) and 0.99 for MoCo v2. For MoCo v2, the queue size, temperature for InfoNCE loss, weight decay were 4096, 1.0, $1.0 \times 10^{-4}$, respectively. We have not employed SyncBN in NCC like BYOL. We note that SyncBN will introduce $1/3$ additional computation costs. Instead, we adopt the shufflingBN in MoCo to avoid the trivial solution of non-contrastive learning.

**Configurations of ImageNet-1K.** For ImageNet-1K, the weight decay and base learning rate were $10^{-4}$ and 0.4, respectively. We employ ResNet-50 as the backbone, train our method for 200 epochs without symetric loss to reduce training time, under the same conditions of PCL (Li et al., 2021a). We include the GaussianBlur to the data augmentations since we use the original image size for training. During evaluation, images are resized to $256 \times 256$ and center cropped. Due to the large computation cost, we run NCC on ImageNet-1K only once.

**Computational cost.** Both BYOL and MoCo use a momentum-updated network $f_{\boldsymbol{\theta}'}'$. The computation cost between them lies on the fact that MoCo needs a large number of negative examples while BYOL uses a large hidden size for the predictor network. Our NCC is built upon BYOL, the main additional computational cost is the k-means clustering procedure. We have implemented the k-means algorithm with k-means++ (Arthur & Vassilvitskii, 2006) initialization using PyTorch to utilize the GPU and accelerate the clustering process. Taking CIFAR-10 and CIFAR-20 datasets for example, it takes only 5 seconds for k-means clustering, leading to only 5 hours of training time. Besides, as suggested in the results of Fig. A2, NCC is robust to the different $r$, so that there is no need to perform k-means for every epoch. The computational cost of the ProtoCL is also small since we build the cluster centers within mini-batch, saying that NCC does not need additional memory to store the cluster centers. Consequently, considering the promising performance improvements, the additional computational cost is relatively affordable.

# D PSEUDOCODE OF NCC

**Algorithm 1** Pseudocode of NCC in a PyTorch-like style

```
1  # f_θ, f'_θ', g_φ: online network, target network, predictor respectively;
2  # K: number of clusters, τ: temperature for ProtoCL loss;
3  # m: momentum, max_epochs: the training epochs;
4  # σ: sampling standard deviation of Gaussian distribution;
5  # λ_pcl: loss weight of protocl loss, λ_pcl = 0 during learning-rate warmup;
6  # r: performs K-means clustering for every r epoch;
7
8  # compute the centers from their representations and pseudo-labels;
9  def compute_centers(z, y):
10     weight = onehot(y).T # KxN;
11     weight = normalize(weight, p=1, dim=1) # ℓ_1-normalization;
12     centers = normalize(weight.mm(z), p=2, dim=1) # ℓ_2-normalization;
13     return centers
14
15 # compute ProtoCL from the cluster centers of two views
16 def compute_protocl_loss(μ_q, μ_k, y):
17     zero_classes = counts(y) # find the empty clusters with indices;
18     # align two views, Kx1;
19     proto_alignment = (μ_q * μ_k).sum(dim=1, keepdim=True) / τ
20     proto_uniformity = μ_q.mm(μ_q.T) / τ # inter-cluster distance;
21     # fill the logits of empty clusters with -10;
22     proto_uniformity = proto_uniformity.fill_(-10, zero_classes, dim=1)
23     # retrive inter-cluster distance for proto uniformity;
24     proto_uniformity = proto_uniformity[off_diagonal_mask] # Kx(K-1);
25     protocl_loss = - proto_alignment + logsumexp(cat([proto_alignment,
       ↪  proto_uniformity], dim=1), dim=1)
26     protocl_loss[zero_classes] = 0 # zero the loss of empty clusters;
27     # neglect the empty clusters;
28     protocl_loss = protocl_loss.sum() / (K - len(zero_classes))
29     return protocl_loss
30
31 # compute both augmented instance alignment and ProtoCL;
32 def forward_loss(im_q, im_k, y):
33     q, k = f_θ(im_q), f'_θ'(im_k) # forward;
34     ins_loss = mse(g_φ(q + σ * randn_like(q)), k) # instance alignment;
35     μ_q = compute_centers(q, y) # compute the centers of two views;
36     μ_k = compute_centers(k, y)
37     protocl_loss = compute_protocl_loss(μ_q, μ_k, y)
38     return ins_loss + protocl_loss * λ_pcl
39
40 θ'.copy_(θ)   # initialize the target network with online network;
41 for epoch in range(max_epochs): # start training;
42     # E-step
43     if epoch % r == 0:
44         # K-means based on target network for every r epochs: N labels
45         pseudo_labels = K-means(loader, f'_θ')
46     # M-step
47     for indices, im in loader: # load a mini-batch x with N samples;
48         im_q, im_k = aug(im) # two randomly data augmentations;
49         y = pseudo_labels[indices] # pseudo-labels in each batch;
50         # symetric loss same as vanilla BYOL: swap inputs of two views;
51         loss = forward_loss(im_q, im_k, y) + forward_loss(im_k, im_q, y)
52         loss /= 2
53         # SGD updating online network and predictor;
54         loss.backward()
55         update(θ, φ)
56         # momentum update target network;
57         θ'.copy_(m*θ' + (1-m)*θ)
```

# E    ADDITIONAL EXPERIMENTAL RESULTS

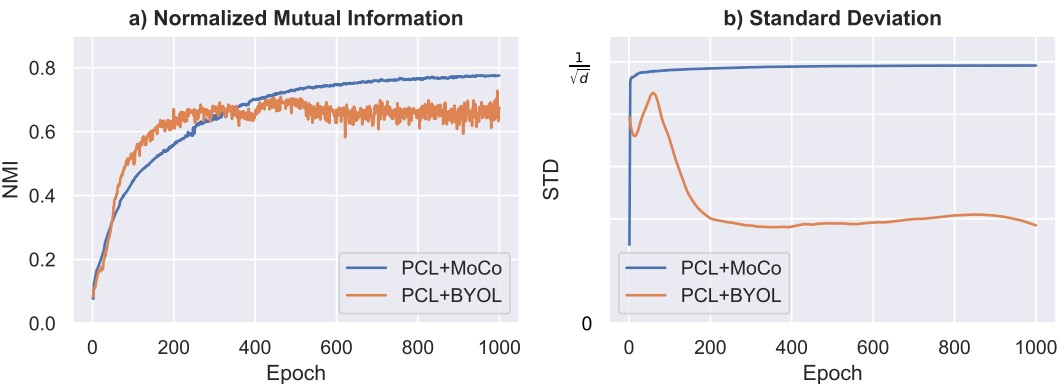

Figure A1:    Visualizations of NMIs and STDs by applying PCL (Li et al., 2021a) to MoCo and BYOL on CIFAR-10. Compared to PCL+BYOL, PCL+MoCo performs more stable during clustering with a more uniform distribution of representations. The decreasing STDs (standard deviation of $\ell_2$-normalized features) also indicate that PCL+BYOL suffers from the representation collapse. This is because PCL can only improve the within-cluster compactness. During training, the fixed prototypes of PCL will also be gradually collapsed without negative examples, making the representations for BYOL collapse at the same time. These results validate our assumptions that BYOL is not robust to additional clustering losses for clustering tasks, since there is no negative example for BYOL to maintain uniform representations to avoid collapse. Our NCC can avoid the class collision issue and representation collapse by the proposed positive sampling strategy to improve the within-cluster compactness and prototypical contrastive loss to maximize the inter-class distance.

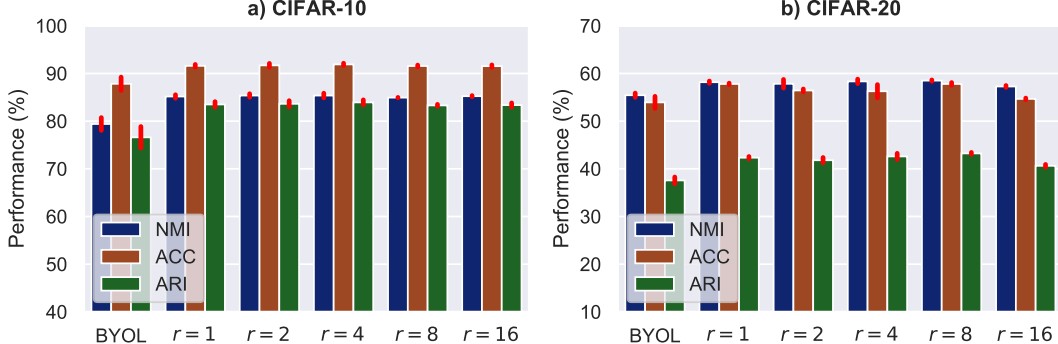

Figure A2: The effect of the hyperparameter $r$. NCC performs K-means clustering for every $r$ epoch. Here we study how different $r$ influences the clustering performance. The results demonstrate NCC is robust to large $r$ and the cluster pseudo-labels, which means it is not necessary to perform clustering for every epoch so that the computation cost can be significantly reduced. In summary, we suggest that $r$ can be set to $[1, 8]$ by considering the datasets and computation resources.

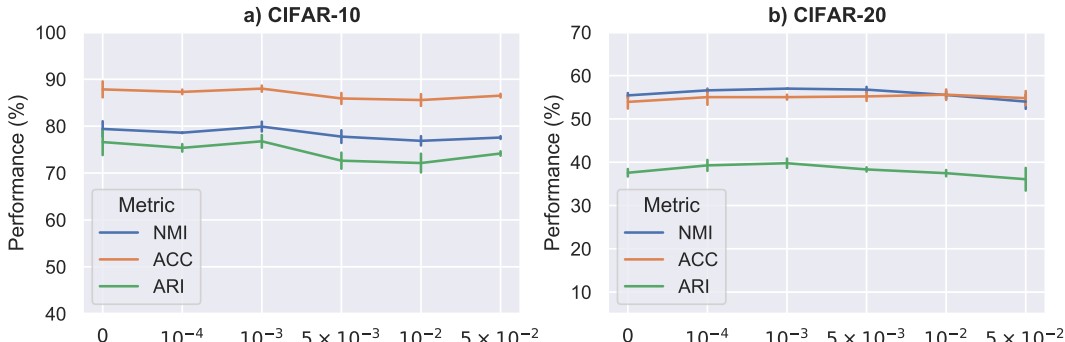

Figure A3: The effect of the hyperparameter $\sigma$ for positive sampling. Taking a look at $\sigma \sim [0, 10^{-3}]$, although introducing the positive sampling into BYOL causes a slight drop on CIFAR-10, the clustering performance becomes more stable as evidenced by the standard deviation. This is because the neighbors of one sample are regarded as positive examples. Besides, the performance for CIFAR-20 has increased over baseline BYOL with the standard deviation reduced. These results indicate that positive sampling can improve the stability of BYOL. However, when $\sigma$ is too large, the performance becomes unstable and drops a lot. It is not surprised since during positive sampling with large $\sigma$, the instances from other clusters could be sampled and regarded as positive examples. Therefore, we suggest setting $\sigma$ to a small value, saying $(0, 10^{-3}]$.

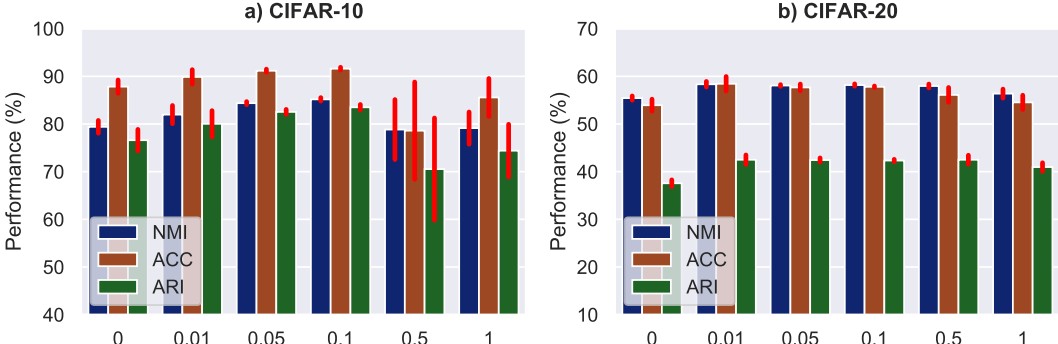

Figure A4: The effect of the hyperparameter $\lambda_{pcl}$ for the ProtoCL. The results suggest that NCC is robust to different loss weights of ProtoCL loss on CIFAR-20. However, the higher loss weight leads to instability on CIFAR-10. The possible reason is that CIFAR-20 is more diverse and has more semantic classes than CIFAR-10 (100 versus 10). Anyway, we suggest that the loss weight for ProtoCL loss can be set to $[0.01, 0.1]$ for different situations, which has demonstrated superior performance on both two datasets.

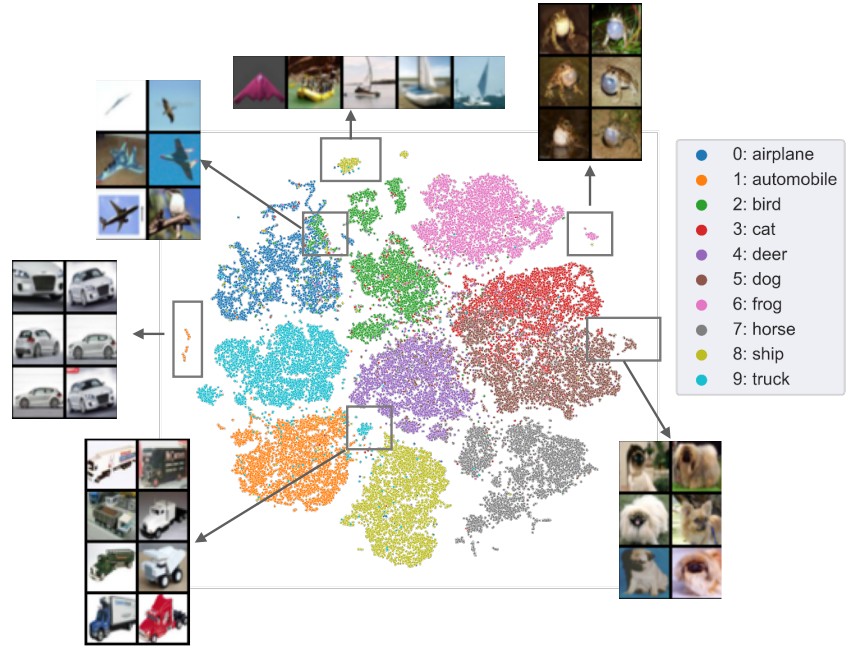

Figure A5: Visualizations for the outlier points produced by the model at 1000-th epoch on CIFAR-10 with t-SNE.

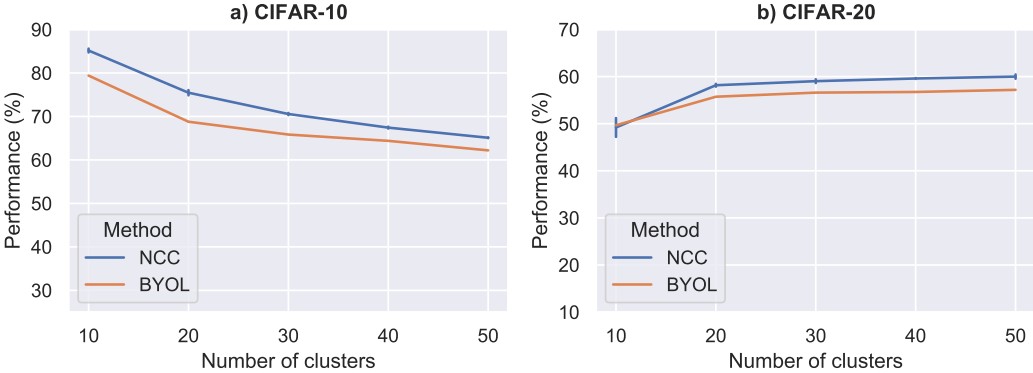

Figure A6: The effect of the predefined number of clusters $K$. We reported NMIs following (Caron et al., 2018). We note that the predefined $K$ of NCC during the training of NCC is the same as the $K$ in k-means clustering process for evaluation. To further demonstrate the influences of over-clustering, we also reported the results of vanilla BYOL during k-means clustering. The results demonstrate that both BYOL and NCC have the same over-clustering behavior on these two datasets; that is, opposite trends on these two datasets. Specifically, over-clustering leads to the performance drop for CIFAR-10, but it leads to performance increase for CIFAR-20. However, our NCC can still produce large improvements over BYOL with the same predefined number of clusters. We note that the opposite results are due to the significant difference between these two datasets. Although having the same number of samples, CIFAR-10 has 10 distinct classes while CIFAR-20 has, in fact, 100 classes but uses 20 super-classes instead. If the representations are well aligned within the same semantic clusters, the over-clustering would try to destroy the structures of the clusters and push the semantically similar examples away, which certainly compromises the clustering performance.

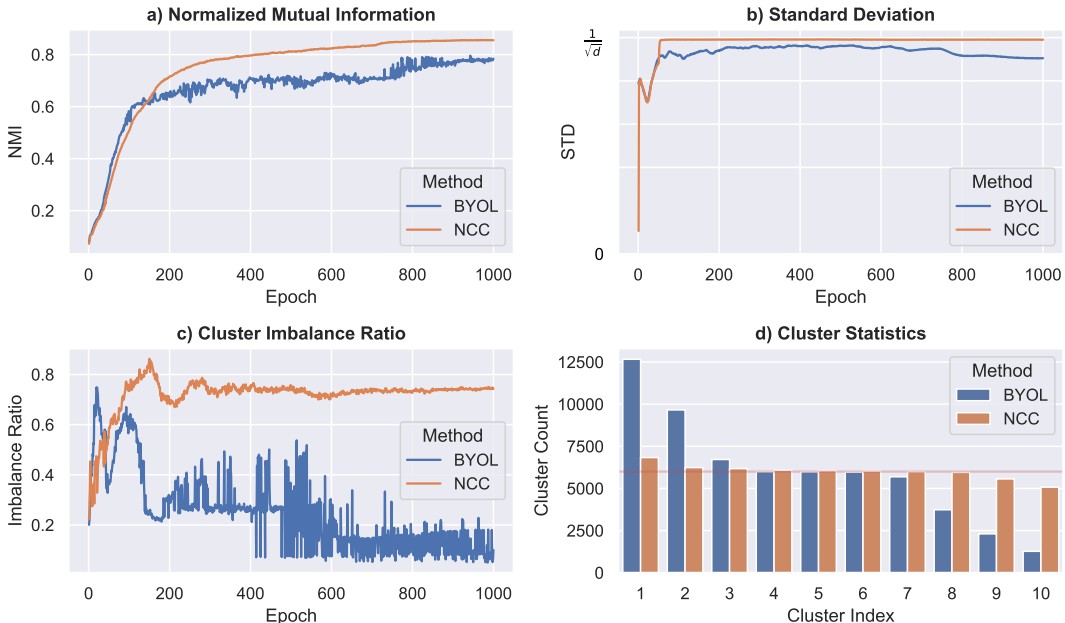

Figure A7: Visualization of training and cluster statistics for BYOL and NCC: a) normalized mutual information (NMI)[1] between the clustering results and ground-truth labels; b) standard deviation (STD)[2] of $\ell_2$-normalized features to evaluate the uniformity, where the higher indicates the more uniform representations; c) cluster imbalance ratio[3] computed by $N_{\min}/N_{\max}$, where $N$ is the number of samples in each class; and d) cluster statistics, or the sorted number of samples in each cluster for the model at 1000-th epoch on CIFAR-10. Taking a look at NMIs and STDs during the training stage in (a) and (b), NCC produces higher NMIs with stable and higher STDs, while BYOL performs unstable with the STDs gradually decreased. The results indicate that NCC yields a more uniform representation space with the samples well-clustered. On the other hand, the uniform representations of NCC can also avoid the collapse of k-means clustering at the same time. As shown in (c), the k-means clustering process of NCC produces more balanced clusters with a higher cluster imbalance ratio. On the contrary, the clusters of BYOL are highly imbalanced, which is consistent with the unstable NMIs and decreasing STDs. Moreover, we visualize the fourth cluster statistic in (d), with the sorted number of samples. Unsurprisingly, the samples for NCC are approximately and equally assigned to different clusters, compared to almost long-tailed assignments of BYOL.

---

[1] NMI is a normalized mutual information between the clustering and ground-truth labels.

[2] Here we visualize the distribution of representations following (Chen & He, 2021). Given the latent vectors $\boldsymbol{z} \sim \mathcal{N}(0, \boldsymbol{I})$, the standard deviation for $\ell_2$-normalized $\boldsymbol{z}' = \boldsymbol{z}/\|\boldsymbol{z}\|_2$ is $\mathrm{std}\left[\boldsymbol{z}'\right] \approx 1/d^{\frac{1}{2}}$, where $d$ is the dimension of $\boldsymbol{z}$. As we discussed in Appendix B, the $\ell_2$-normalized standard Gaussian distribution can be approximately seen as the uniform vMF distribution on the hypersphere. Therefore, a higher standard deviation close to $1/d^{\frac{1}{2}}$ indicates more uniform representations.

[3] Same as the coefficient to control the data imbalance for the long-tailed datasets in Table. A4, a higher value indicates more balanced clusters.

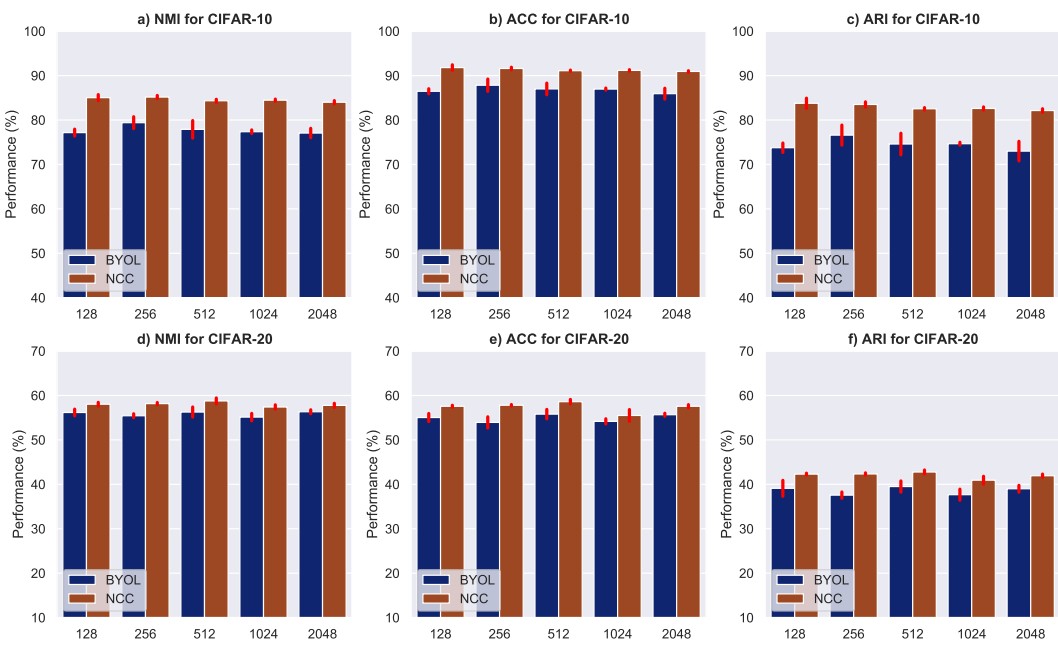

Figure A8: The effect of the different projection dimension. NCC achieves consistent and significant performance improvement over BYOL regardless of different projection dimension.

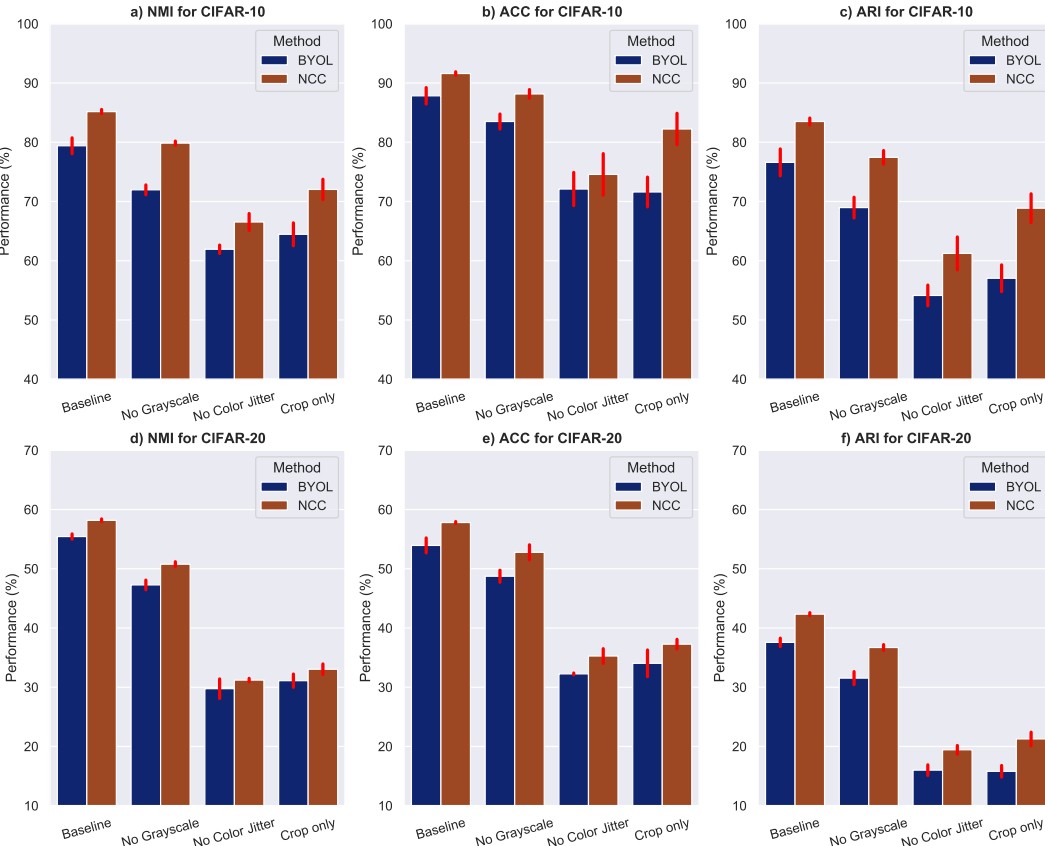

Figure A9: The effect of the different data augmentation. It is not surprised to see the performance drops for both BYOL and NCC when removing some data augmentations. On the contrary, the clustering results suggest that NCC still performs more stable and is robust to data augmentations.

Table A1: Clustering results (%) for fair comparisons. We train NCC to demonstrate its effectiveness for fair comparisons with the following settings: 1) we exclude test set from the whole dataset; and 2) we use an original image size (224) for ImageNet-10 and ImageNet-Dogs. All results were trained with ResNet-34. There is no clear margin for CIFAR-10 and CIFAR-20 datasets with different splits, while significant improvements can be observed for ImageNet-10 and ImageNet-Dogs datasets. Considering that NCC has already achieved state-of-the-art performance against previous work in Table 2, these results further demonstrate the superiority of NCC.

| | NMI | ACC | ARI | NMI | ACC | ARI |
|---|---|---|---|---|---|---|
| | **CIFAR-10** | | | **CIFAR-20** | | |
| Baseline (train+test) | **88.6**±1.0 | **94.3**±0.6 | **88.4**±1.1 | 60.6±0.3 | 61.4±1.1 | 45.1±0.1 |
| Exclude test set | 88.3±0.2 | 94.2±0.2 | 88.1±0.3 | **61.2**±0.9 | **61.5**±0.9 | **45.9**±0.5 |
| | **ImageNet-10** | | | **ImageNet-Dogs** | | |
| Baseline (96) | 89.6±0.2 | 95.6±0.0 | 90.6±0.1 | 69.2±0.3 | 74.5±0.1 | 62.7±0.1 |
| Large image size (224) | **90.8**±0.4 | **96.2**±0.1 | **91.8**±0.3 | **73.7**±0.2 | **77.5**±0.1 | **67.5**±0.1 |

Table A2: Clustering results (%) on the subsets of ImageNet. We strictly follow the settings in (Van Gansbeke et al., 2020): we have adopted the same 50, 100, and 200 classes from ImageNet, clustered on the training set and tested on the validation set. We have used the same experimental settings as the other benchmarked datasets and trained NCC with ResNet-50 for 300 epochs. We note that SCAN has used the pre-trained model of MoCo trained on the full ImageNet for 800 epochs. The results are directly referred from their published paper including k-means with pre-trained MoCo[1], SCAN after the clustering step[2], and SCAN after the self-labeling step[3]. With much fewer training epochs and training data, NCC still produces better performance with a clear margin, demonstrating the superiority of NCC.

| ImageNet | **50 Classes** | | **100 Classes** | | **200 Classes** | |
|---|---|---|---|---|---|---|
| Method | NMI | ARI | NMI | ARI | NMI | ARI |
| k-means[1] | 77.5 | 57.9 | 76.1 | 50.8 | 75.5 | 43.2 |
| SCAN[2] | 80.5 | 63.5 | 78.7 | 54.4 | 75.7 | 44.1 |
| SCAN[3] | 82.2 | 66.1 | 80.8 | 57.6 | 77.2 | 47.0 |
| NCC (Ours) | **82.8** | **69.1** | **83.5** | **63.5** | **80.6** | **53.8** |

Table A3: Clustering results (%) on Tiny-ImageNet (Le & Yang, 2015). Tiny-ImageNet is also a subset of ImageNet, with a total of 200 classes and 500 images for each class. We trained NCC with same settings in Table 2 except for ResNet-18 and the image size $96 \times 96$. Even though with a smaller backbone and image size (CC used ResNet-34 and the image size $224 \times 224$), our NCC still produces promising improvements over previous state-of-the-art methods.

| Dataset | **Tiny-ImageNet** | | |
|---|---|---|---|
| Method | NMI | ACC | ARI |
| k-means (Lloyd, 1982) | 6.5 | 2.5 | 0.5 |
| SC (Zelnik-manor & Perona, 2005) | 6.3 | 2.2 | 0.4 |
| AE (Bengio et al., 2007) | 13.1 | 4.1 | 0.7 |
| VAE (Kingma & Welling, 2014) | 11.3 | 3.6 | 0.6 |
| JULE (Yang et al., 2016) | 10.2 | 3.3 | 0.6 |
| DEC (Xie et al., 2016) | 11.5 | 3.7 | 0.7 |
| DAC (Chang et al., 2017) | 19.0 | 6.6 | 1.7 |
| DCCM (Wu et al., 2019) | 22.4 | 10.8 | 3.8 |
| PICA (Huang et al., 2020) | 27.7 | 9.8 | 4.0 |
| CC (Li et al., 2021b) | 34.0 | 14.0 | 7.1 |
| GCC (Zhong et al., 2021) | 34.7 | 13.8 | 7.5 |
| NCC (Ours) | **40.5** | **25.6** | **14.3** |

Table A4: Clustering results (%) on long-tailed datasets of different self-supervised learning frameworks and our proposed NCC. We built the long-tailed version of CIFAR-10 and CIFAR-20, termed CIFAR-10-LT and CIFAR-20-LT using the codes of (Tang et al., 2020), which follows (Zhou et al., 2020; Cao et al., 2019). Specifically, they were built upon the training datasets under the control of data imbalance ratio $N_{min}/N_{max} = 0.1$ for the data distribution, where $N$ is the number of samples in each class. The samples in the long-tailed datasets are almost all in the head of distributions. MoCo cannot handle this problem well due to the class collision issue, as a results, the samples in the head will be pushed away and the ones in the tail will be mixed together. The rest three methods do not need the negative examples so that they outperform MoCo v2 by a large margin. By introducing the positive sampling and ProtoCL, we can further boost and stabilize the performance of vanilla BYOL.

| Methods | CIFAR-10-LT | | | CIFAR-20-LT | | |
| --- | --- | --- | --- | --- | --- | --- |
| | NMI | ACC | ARI | NMI | ACC | ARI |
| MoCo v2 | 46.7±0.1 | 33.4±0.3 | 27.7±0.0 | 31.2±0.3 | 28.2±0.2 | 16.1±0.3 |
| BYOL | 51.6±1.0 | 41.3±0.4 | 30.8±0.4 | 41.9±0.4 | 34.6±0.5 | 22.3±1.0 |
| NCC | **55.3**±0.4 | **43.9**±0.1 | **36.3**±0.3 | **44.6**±0.2 | **39.0**±0.7 | **27.3**±0.2 |
| w/o ProtoCL | 53.1±0.7 | 42.7±0.4 | 31.6±0.8 | 43.4±0.8 | 35.1±0.6 | 24.0±0.1 |

Table A5: Clustering results (%) on different ResNet architectures. With the deeper ResNet networks, there are little performance gain and even drops for MoCo v2 and BYOL. On the contrary, NCC achieves significant improvement with small standard deviation for clustering, demonstrating its superior stability and performance against the baseline MoCo v2 and BYOL. The best and second best results are shown in bold and underline, respectively.

| Backbone | Method | Dataset | NMI | ACC | ARI | Dataset | NMI | ACC | ARI |
| --- | --- | --- | --- | --- | --- | --- | --- | --- | --- |
| ResNet-18 | MoCo | | 76.9±0.2 | 84.9±0.3 | 72.4±0.5 | | 49.2±0.1 | 48.0±0.2 | 32.1±0.0 |
| | BYOL | | 79.4±1.7 | 87.8±1.7 | 76.6±2.8 | | 55.5±0.6 | 53.9±1.6 | 37.6±0.9 |
| | Ours | | **85.1**±0.5 | **91.6**±0.4 | **83.5**±0.7 | | **58.2**±0.3 | **57.8**±0.2 | **42.3**±0.3 |
| ResNet-34 | MoCo | CIFAR-10 | 79.4±0.0 | 86.9±0.0 | 75.6±0.0 | CIFAR-20 | 51.3±0.0 | 49.2±0.6 | 34.3±0.3 |
| | BYOL | | 81.7±1.0 | 89.4±0.6 | 79.0±1.0 | | 55.9±0.3 | 56.9±1.8 | 39.3±0.2 |
| | Ours | | **88.6**±1.0 | **94.3**±0.6 | **88.4**±1.1 | | **60.6**±0.3 | **61.4**±1.1 | **45.1**±0.1 |
| ResNet-50 | MoCo | | 79.9±0.1 | 87.1±0.1 | 76.0±0.1 | | 53.3±0.0 | 52.9±0.1 | 36.7±0.1 |
| | BYOL | | 80.0±0.4 | 88.1±0.9 | 76.3±1.8 | | 54.1±1.4 | 53.1±2.4 | 36.9±2.9 |
| | Ours | | **89.4**±0.5 | **94.7**±0.4 | **89.1**±0.7 | | **63.2**±1.0 | **61.3**±1.2 | **47.1**±1.4 |
| ResNet-18 | MoCo | | 83.3±0.1 | 90.4±0.1 | 81.6±0.1 | | 38.7±0.5 | 36.7±0.5 | 23.9±0.6 |
| | BYOL | | 87.2±0.6 | 94.2±0.5 | 87.6±1.1 | | 59.6±0.8 | 66.1±1.1 | 49.8±1.1 |
| | Ours | | **89.3**±0.4 | **95.4**±0.1 | **89.7**±0.8 | | **64.4**±1.3 | **69.6**±2.2 | **56.1**±2.3 |
| ResNet-34 | MoCo | ImageNet-10 | 84.2±0.1 | 90.7±0.1 | 82.3±0.1 | ImageNet-Dogs | 41.6±1.2 | 39.0±1.5 | 26.8±1.7 |
| | BYOL | | 86.6±0.2 | 93.9±0.1 | 87.2±0.2 | | 63.5±2.2 | 69.4±3.0 | 54.8±2.9 |
| | Ours | | **89.6**±0.2 | **95.6**±0.0 | **90.6**±0.1 | | **69.2**±0.3 | **74.5**±0.1 | **62.7**±0.1 |
| ResNet-50 | MoCo | | 82.6±0.1 | 90.0±0.0 | 80.9±0.0 | | 40.8±1.0 | 38.6±1.1 | 25.4±0.7 |
| | BYOL | | 88.7±1.8 | 95.2±0.9 | 89.7±1.9 | | 66.2±0.2 | 71.9±1.2 | 58.1±1.2 |
| | Ours | | **91.5**±0.2 | **96.5**±0.1 | **92.5**±0.2 | | **72.0**±0.2 | **76.3**±0.1 | **65.7**±0.1 |

