# OpenReview forum: "Exploring Non-Contrastive Representation Learning for Deep Clustering"
_ICLR.cc/2022/Conference — ICLR 2022 Submitted_

### Official Review · Reviewer_nh13 · 2021-11-02

**Correctness:** 3
**Technical Novelty And Significance:** 4
**Empirical Novelty And Significance:** 3
**Recommendation:** 8
**Confidence:** 3

**Main Review:**

## Strength

### Good performance

The paper demonstrates a clear improvement over existing methods on various datasets. It is unfortunate that the method is not tested on the same settings as PCL in Table 2, but Table 3 does seem to show that there is a large enough gap between the two.

### Well motivated

The motivation for the paper is really on the fact that negatives harm, and the paper shows how to effectively avoid this from happening. Have your clusters well spread, but don't ask for individual samples to be further away, simply ask them to be closer to each center. And by enforcing the centres to be spread, you naturally avoid asking two different samples close by to be further away, which may not be nice.

## Weaknesses

### Missing PCL in Table 2

PCL is perhaps the exact antithesis of the proposed method. The method already shows to work better than PCL in Table 3, so why not include it in Table 2 to make it VERY clear that the proposed way is the way to go compared to PCL? This is personally a must-have experiment for me.

### Ablation study

The ablation study can be improved by perhaps also showing how the ProtoNCE's formulation of negative losses harms performance. For example, what happens if you simply add that term to the NCC formulation?

Also, from the ablation study, it is slightly questionable whether the alignment loss is necessary. In fact, shouldn't the positive sampling also do the same thing as the alignment loss?

### Presentation

The presentation can be improved. The paper is not too hard to follow but could be presented in a more top-down way. This might be a presentation style issue, and it does come to personal taste, so please do take this into account only as a suggestion.

The abstract does not read well. There are multiple novel terms introduced, "a positive sampling strategy" and "prototypical contrastive loss" that are not explained, which makes it hard to follow. For example,

> First, we propose a positive sampling strategy to align one augmented view of instance with the neighbors of another view so that we can avoid the class collision issue caused by the negative examples and hence improve the within-cluster compactness.

Can be simply written as

> First, we relate between neighbouring instances in the learned latent space to improve within-cluster compactness leading to less collision of non-related classes.

Prototypical contrastive loss also can be presented in a similar way, without necessarily introducing the term. In fact, I would personally argue that it is easier to understand the method the other way around. The main idea of the paper seems to be that negative samples are risky, therefore, let's cluster samples in the latent space to form prototypical clusters, and apply contrastive loss only on the cluster centres and not on the features themselves, unlike in the case of ProtoNCE where all samples are optimized. This however risks clusters becoming meaningless, hence the cluster grouping loss which maximizes the likelihood of each sample being from a nearby cluster with a Gaussian assumption.

This problem persists also in the introduction, as the reader is unaware of what the paper means by "positive sampling" until Section 3.2, and Prototypical contrastive loss until 3.3. This problem is exacerbated since it is unclear what 3.2 and 3.3 are trying to achieve until you reach 3.4, which provides the picture in which all of these fit in place.


**Summary Of The Paper:**

The paper proposes that negative samples are risky, therefore, let's cluster samples in the latent space to form prototypical clusters, and apply contrastive loss only on the cluster centres and not on the features themselves, unlike in the case of ProtoNCE where all samples are optimized. This however risks clusters becoming meaningless, hence the cluster grouping loss which maximizes the likelihood of each sample being from a nearby cluster with a Gaussian assumption. The method is tested on multiple datasets outperforming all compared methods.

**Summary Of The Review:**

The paper shows promising results, with a great motivation. However, there are some issues that can be easily fixed to greatly improve the quality of the paper. Specifically, ProtoNCE, or PCL, should be included in Table 2, and an additional baseline that empirically shows the negative loss harming performance would be great. Presentation could also be improved to be more clear.

---

> ### Author Response · Authors · 2021-11-18
> **Response to Reviewer nh13 (1/2)**
>
> Thank you for the encouraging and valuable feedback.
>
> **`1. "Missing PCL in Table 2"`**
>
> We would like to note that the reason why we had PCL in Table 3 but not in Table 2 is that PCL is focused on self-supervised representation learning and only reported clustering on the ImageNet-1k dataset. Before we move to the results of PCL on more datasets, we would like to clarify the conceptual difference between PCL and our NCC.
>
> 1. **Our NCC can avoid class collision issue while PCL cannot.** NCC is based on BYOL that does not require negative examples for representation learning while PCL is based on instance-wise contrastive loss that requires a number of negative examples for representation learning, inevitably leading to class collision issue.
>
> 2. **The proposed ProtoCL in NCC is conceptually different from the ProtoNCE in PCL.** ProtoCL is to maximize the inter-cluster distance to form a uniformly distributed space while ProtoNCE is to minimize the instance-to-cluster distance to improve the within-cluster compactness. The within-cluster compactness of NCC is improved by the proposed positive sampling strategy.
>
> 3. **Pure ProtoCL can work well for deep clustering while ProtoNCE requires another InfoNCE to form uniformly distributed space**. This is a direct result of the different designs of the losses. ProtoCL can maximize the inter-cluster distance to form a uniformly distributed space while ProtoNCE suffers from collapse without the help of another InfoNCE to form such a space.
>
> Next, we conducted these comparisons using the official source code of PCL, where the training details can be found in the revised manuscript. The results are shown below in the form of NMI/ACC/ARI for the five benchmark datasets. The results show that our NCC consistently outperforms PCL on all datasets by a clear margin, demonstrating the superiority of NCC.
>
> | Method |          CIFAR-10          |          CIFAR-20          |           STL-10           |        ImageNet-10         |       ImageNet-Dogs        |
> | ------ | :------------------------: | :------------------------: | :------------------------: | :------------------------: | :------------------------: |
> | PCL    |       80.2/87.4/76.6       |       52.8/52.6/36.3       |       71.8/41.0/67.0       |       84.1/90.7/82.2       |       44.0/41.2/29.9       |
> | NCC    | **88.6**/**94.3**/**88.4** | **60.6**/**61.4**/**45.1** | **75.8**/**86.7**/**73.7** | **89.6**/**95.6**/**90.6** | **69.2**/**74.5**/**62.7** |
>
> **`2. "Also, from the ablation study, it is slightly questionable whether the alignment loss is necessary. In fact, shouldn't the positive sampling also do the same thing as the alignment loss?"`**
>
> First, we would like to note that the prototypical alignment loss is actually a part of the prototypical contrastive loss. We separate them for discussion as prototypical alignment loss and prototypical uniformity loss have different meanings in BYOL framework.
>
> Although the only prototypical uniformity term can outperform ProtoCL loss on CIFAR-10, the prototypical alignment term is necessary to improve the results on CIFAR-20. We believe that the prototypical alignment term is essential to stabilize the training process as it can align two data augmentation of online and target networks;  this is also essential as ProtoCL computes the cluster centers within a mini-batch.
>
> Positive sampling aims to augment the instance alignment loss to improve the within-cluster compactness by encouraging the neighbors as positive examples. It neither helps to maximize inter-cluster distance nor aligns the data augmentations of the online and target networks. Therefore, positive sampling does not do the same thing as the alignment loss.
>
> In addition, as we cannot cover all clusters with a small batch size and a larger number of clusters, the training would be unstable as evidenced by the large standard deviations on CIFAR-20. In a sense, prototypical alignment loss performs as an anchor from the target network to guide the training of the online network. In summary, the prototypical alignment loss would be necessary when there is a large number of clusters.
>
>
> **Continued below...**

---

> > ### Author Response · Authors · 2021-11-18
> > **Response to Reviewer nh13 (2/2)**
> >
> > **...continued from Response (1/2) above**
> >
> > **`3. "The ablation study can be improved by perhaps also showing how the ProtoNCE's formulation of negative losses harms performance. For example, what happens if you simply add that term to the NCC formulation?"`**
> >
> > Per your suggestion, we conducted the ablation study that adds ProtoNCE into our NCC formulation by replacing our ProtoCL with ProtoNCE, which is equal to BYOL+PCL. The results are shown in the following table. The results demonstrate that BYOL is not robust to additional losses for clustering tasks, as directly applying PCL to BYOL will significantly harm the clustering performance and make the model unstable.
> >
> > | **CIFAR-10** |       NMI        |       ACC        |       ARI        | **CIFAR-20** |       NMI        |       ACC        |       ARI        |
> > | ------------ | :--------------: | :--------------: | :--------------: | ------------ | :--------------: | :--------------: | :--------------: |
> > | PCL          |   77.6$\pm$0.1   |   85.5$\pm$0.1   |   73.4$\pm$0.0   | PCL          |   50.0$\pm$0.3   |   48.6$\pm$0.7   |   32.7$\pm$0.4   |
> > | BYOL         |   79.4$\pm$1.7   |   87.8$\pm$1.7   |   76.6$\pm$2.8   | BYOL         |   55.5$\pm$0.6   |   53.9$\pm$1.6   |   37.6$\pm$0.9   |
> > | BYOL+PCL     |   74.4$\pm$2.3   |   85.3$\pm$0.9   |   71.4$\pm$1.4   | BYOL+PCL     |   49.7$\pm$0.7   |   46.9$\pm$0.7   |   27.8$\pm$1.5   |
> > | NCC (ours)   | **85.1**$\pm$0.5 | **91.6**$\pm$0.4 | **83.5**$\pm$0.7 | NCC (ours)   | **58.2**$\pm$0.3 | **57.8**$\pm$0.2 | **42.3**$\pm$0.3 |
> >
> > We list the Normalized Mutual Information and Standard Deviation (STD, $\times \sqrt{d}$ like SimSiam where $d$ is the feature dimension) of the $\ell_2$-normalized features in the following table. The high STDs indicate more uniform representations on the hypersphere. Compared to the MoCo-based PCL and vanilla BYOL, applying ProtoNCE loss to BYOL would decrease the clustering performance since the representations collapsed (see lowest STDs).
> >
> > | Method   | Normalized Mutual Information | Standard Deviation |
> > | :------- | :---------------------------: | :----------------: |
> > | PCL+MoCo |         77.6$\pm$0.1          |       0.986        |
> > | PCL+BYOL |         74.4$\pm$2.3          |       0.375        |
> > | NCC      |         **85.1**$\pm$0.5          |       **0.990**        |
> >
> > We think that the possible reason is that PCL only improves the within-cluster compactness. During training, the fixed prototypes of PCL will also be gradually collapsed without negative examples to form a uniformly distributed space, making the representations for BYOL collapse at the same time. These results validate our assumptions that BYOL is not robust to additional clustering losses for clustering tasks, since there is no negative example for BYOL to maintain uniform representations to avoid collapse. Our NCC can avoid the class collision issue and representation collapse by the proposed positive sampling strategy to improve the within-cluster compactness and prototypical contrastive loss to maximize the inter-class distance.
> >
> >
> > **`4. "The presentation can be improved"`**
> >
> > We highly appreciate your suggestions on how to improve the presentation of this paper and improve the quality. We tried our best to follow your suggestion in the revised manuscript.

---

> > > ### Comment · Reviewer_nh13 · 2021-11-19
> > > **Delay in response**
> > >
> > > Hi Authors!  Thank you so much for the detailed response. I've been struck hard by the CVPR deadline, and am recovering, but I want to make sure that I look at your response & the paper once more with a clear brain. I think I'll be able to post a reply tomorrow, but just wanted to let you know that you are not being unheard!

---

> ### Comment · Reviewer_nh13 · 2021-11-20
> **Response to authors' comments**
>
> I would like to first thank the authors for their **extensive** effort in the response. All weak points of the paper that I have pointed out have been very well addressed. I especially appreciate the new ablation studies and the comparisons that reveal that my worries were just unnecessary concerns. Still, I am very happy to have helped improve the quality of the paper.
>
> Thanks!

---

> > ### Author Response · Authors · 2021-11-20
> > **Response to Reviewer nh13**
> >
> > Dear Reviewer nh13,
> >
> > Thank you very much again for providing constructive and helpful suggestions, which have significantly helped improve the quality, impact, and clarity of this paper.
> >
> > All the best,
> >
> > Authors

---

### Official Review · Reviewer_dith · 2021-11-02

**Correctness:** 3
**Technical Novelty And Significance:** 2
**Empirical Novelty And Significance:** 3
**Recommendation:** 6
**Confidence:** 4

**Main Review:**

### Strengths
* The comparison with the prior works and the motivation of the proposed method are well described and reasonable.
* The efficacy of the proposed methods are well proven with the experimental results.
* A lot of experimental results on hyperparameters search show that the proposed method is quite robust.


### Weaknesses
* Class-collision issue arose from k-means clustering

    I think it's kind of a chicken and egg situation, cause if the space is not well uniformly distributed, the result of the k-means clustering would be quite degenerated which would result in a severe class collapse issue.
Have you checked the k-means clustering results with the ground truth class labels?

* Lack of analysis on the over-clustering results

    Based on the results in Table 4 and Figure A5, it seems that the pre-defined value of the number of clusters, K, has the largest impact on the performance (more than 10% in terms of accuracy score). It is quite intuitive to think that the more the clusters are set, the better the classification performance (cause there are more exemplars to infer the class), but the results are the opposite of this idea for the CIFAR-10 dataset. Since it is the factor that impacts the performance most, I think more elaboration of this result is required.

**Summary Of The Paper:**

This paper proposes a novel method of self-supervised representation learning.
To circumvent the class collision issue arose from building a large set of negative samples in contrastive SSL based methods, it is built from non-contrastive SSL methods, such as BYOL.
The goal is to handle the weaknesses of non-contrastive SSL methods, training instability and representation collapse.
The two proposed methods for alleviating the two factors are i) augmented positive sampling and ii) optimizing uniformity of representation space via prototypical cluster features computed from k-means clustering.


**Summary Of The Review:**

I think this paper is well written in that the motivation, the description of methodology and the analysis of the results are all reasonable.
Though I feel more details are needed for some factors that i described in the main review section, I think this work is good since the proposed method has pointed out the weaknesses of the previous methods and improved it experimentally.

---

> ### Author Response · Authors · 2021-11-18
> **Response to Reviewer dith (1/2)**
>
> Thank you for the encouraging and valuable feedback.
>
> **`1. "Class-collision issue arose from k-means clustering" `**
>
> This is a very insightful and interesting question. We agree with you that if the space is not well uniformly distributed, the result of the k-means clustering would quite degenerate which would result in a severe class collapse issue. In this paper, we mainly considered the class collision issue that arose from the representation learning.
>
> Our NCC performs the clustering and representation learning simultaneously, where the proposed ProtoCL tries to make cluster centers uniformly distributed on the hypersphere and the proposed positive sampling strategy can improve the within-cluster compactness by sampling the neighbors of one example as positive examples. NCC can maintain a uniform representation space by only maximizing the inter-cluster distance, which would prevent the k-means from collapsing. On the other hand, NCC iteratively estimates the pseudo-labels via spherical k-means based on the target network and optimizes the online network via the proposed losses. As the training goes, the k-means clustering can assign the semantic similar samples into the same clusters and make the representations better for clustering.
>
> Experimentally, we first show the training and cluster statistics for BYOL and NCC on CIFAR-10, including
>
> 1) normalized mutual information (NMI) between the clustering and ground-truth labels as requested;
>
> 2) standard deviation (STD, $\times \sqrt{d}$ like SimSiam where $d$ is the feature dimension) of $\ell_2$-normalized features, where the higher STD indicates the more uniform representations;
>
> 3) cluster imbalance ratio (CIR) computed by $N_{\text{min}}/N_{\text{max}}$, where $N$ is the number of samples in each class; and
>
> 4) cluster statistics or the sorted number of samples in each cluster for the model at 1000-th epoch.
>
> We reported the first three statistics in the following table.
>
> **Visualization of training (NMI/STD/CIR)**
>
> | Epoch |              400              |              600              |              800              |             1000              |
> | ----- | :---------------------------: | :---------------------------: | :---------------------------: | :---------------------------: |
> | BYOL  |       0.708/0.961/0.270       |       0.720/0.956/0.075       |       0.746/0.917/0.131       |       0.785/0.905/0.100       |
> | NCC   | **0.797**/**0.992**/**0.743** | **0.823**/**0.991**/**0.721** | **0.851**/**0.991**/**0.729** | **0.855**/**0.990**/**0.743** |
>
> Taking a look at NMIs and STDs during the training stage, NCC produces higher NMIs with stable and higher STDs, while BYOL performs unstable with the STDs gradually decreased. The results indicate that NCC yields a more uniform representation space (see stable STDs across epochs) with the samples well-clustered (see higher NMIs). On the other hand, the uniform representations of NCC can also avoid the collapse of k-means clustering at the same time. As shown in CIR, the k-means clustering process of NCC produces more balanced clusters with a higher cluster imbalance ratio. On the contrary, the clusters of BYOL are highly imbalanced, which is consistent with the unstable NMIs and decreasing STDs.
>
> Second, we listed the fourth cluster statistics, with the sorted number of samples, in the following table. Unsurprisingly, the samples for NCC are approximately and equally assigned to different clusters, compared to almost long-tailed assignments of BYOL.
>
> **Cluster statistics (sorted by the number of samples, 6000 per cluster)**
>
> | Cluster Index |     0 |    1 |    2 |    3 |    4 |    5 |    6 |    7 |    8 |    9 |
> | ------------- | ----: | ---: | ---: | ---: | ---: | ---: | ---: | ---: | ---: | ---: |
> | BYOL          | 12665 | 9659 | 6719 | 5995 | 5986 | 5968 | 5693 | 3736 | 2308 | 1271 |
> | NCC           |  6834 | 6240 | 6178 | 6077 | 6050 | 6029 | 5994 | 5956 | 5566 | 5076 |
>
>
>
> At last, we also conducted experiments in our submitted manuscript on the long-tailed datasets under the control of data imbalance ratio $N_{\text{min}}/N_{\text{max}}=0.1$. The results indicate that NCC can be easily adapted into the imbalance datasets without the k-means clustering collapse.
>
> In summary, NCC can avoid class collision issue for representation learning with the help of BYOL and does not observe the class collision issue for k-meaning clustering with the help of the proposed ProtoCL in the deep clustering paradigm. The experimental results also validate that these two issues can be significantly alleviated in NCC compared to BYOL.
>
> **Continued below...**

---

> > ### Author Response · Authors · 2021-11-18
> > **Response to Reviewer dith (2/2)**
> >
> > **...continued from Response (1/2) above**
> >
> > **`2. "Lack of analysis on the over-clustering results"`**
> >
> > First, we would like to note that the opposite results are due to the significant difference between these two datasets. Although having the same number of samples, CIFAR-10 has 10 distinct classes while CIFAR-20 has, in fact, 100 classes but uses 20 super-classes instead. If the representations are well aligned within the same semantic clusters, the over-clustering would try to destroy the structures of the clusters and push the semantically similar examples away, which certainly compromises the clustering performance.
> >
> > Second, to further demonstrate the influences of over-clustering, we also reported the NMIs of vanilla BYOL during k-means clustering under a different number of clusters, as shown in the following table.
> >
> > | **CIFAR-10** |    10    |    20    |    30    |    40    | 50       | **CIFAR-20** |    20    |    30    |    40    |    50    |
> > | :----------: | :------: | :------: | :------: | :------: | -------- | :----------: |  :------: | :------: | :------: | :------: |
> > |     BYOL     |   79.4   |   68.8   |   65.8   |   64.4   | 62.2     | BYOL              |   55.7   |   56.6   |   56.7   |   57.2   |
> > |     NCC      | **85.2** | **75.5** | **70.6** | **67.4** | **65.1** | NCC            | **58.2** | **59.0** | **59.6** | **60.0** |
> >
> > We note that the predefined $K$ of NCC is consistent for both the training and k-means clustering process. The results demonstrate that both BYOL and NCC have the same over-clustering behavior on these two datasets; that is, opposite changes on these two datasets. Specifically, over-clustering leads to the performance drop for CIFAR-10, but it leads to performance increase for CIFAR-20. However, our NCC can still produce large improvements over BYOL with the same predefined number of clusters.

---

> > ### Comment · Reviewer_dith · 2021-11-20
> > **Response to the authors' rebuttal comments**
> >
> > Dear Authors,
> >
> > I appreciate for your effort for clarifying the points I was curious about.
> > The analysis on the results of k-means clustering is impressive.
> > I think it really strengthens your claim that the proposed method would make the embedding space to have a distinctive cluster set.
> > I had fun while reviewing this paper since I agree that handling the class collision issue is important for robust cluster learning.
> > Thanks again for this additional materials.
> >
> > Sincerely, \
> > Reviewer dith

---

> > > ### Author Response · Authors · 2021-11-20
> > > **Response to Reviewer dith**
> > >
> > > Dear Reviewer dith,
> > >
> > > We appreciate your insightful comments, which have helped improve the quality and impact of this work. We completely agree with you that handling the class collision issue is important for robust cluster learning and hope this study will attract the community’s attention to the non-contrastive representation learning methods for deep clustering. It is with great pleasure that we received your positive feedback and constructive suggestions. Many thanks!
> > >
> > > All the best,
> > >
> > > Authors

---

### Official Review · Reviewer_Na7g · 2021-11-02

**Correctness:** 4
**Technical Novelty And Significance:** 2
**Empirical Novelty And Significance:** 2
**Recommendation:** 3
**Confidence:** 5

**Main Review:**

Pros:

(1) The writing and organization are good, which makes the paper easy to read and follow.

(2) The authors conduct extensive experiments, including the ImageNet to show the superiority. The results are good.

Cons:

(1) My main concern lies in the novelty, which is very limited. This paper simply combines two existing unsupervised learning methods, BYOL and PCL, and then applies it to the clustering task. The class collision issue for negative samples contrastive learning methods is well addressed by BYOL instead of this paper. The difference between the proposed ProtoCL and PCL comes from the BYOL. The EM framework is also presented by the PCL. Besides, the neighbor examples based positive sampling strategy has been well investigated in both unsupervised feature learning and deep clustering areas [1,2]. Therefore, the only contribution is the combination and application to the clustering task, which is obviously insufficient.

(2) Two important references [1,2] are missing. These two methods thoroughly study the positive sampling strategy for clustering and unsupervised feature learning to handle the issue of inaccurate negative samples. Therefore, the positive sampling strategy in NCC is not new. Please also compare the results with these methods.

(3) Though the reported results are very high, the comparison might be unfair. Some existing contrastive learning based clustering methods, such as CC and GCC [1], can also adopted the framework of BYOL and PCL to improve the results. Therefore, the results might be unconvincing.

[1] Graph Contrastive Clustering, ICCV 2021
[2] Weakly Supervised Contrastive Learning, ICCV 2021

**Summary Of The Paper:**

For deep clustering, this paper explores the non-contrastive representation learning based on BYOL to handle the issue of the class collision caused by inaccurate negative samples. ProtoCL is proposed to encourage prototypical alignment between two augmented views and prototypical uniformity, hence maximizing the inter-cluster distance. Experiments on various datasets demonstrate the superiority of the proposed method.

**Summary Of The Review:**

This paper proposes the non-contrastive representation learning method for deep clustering based on BYOL. It is a combination of two existing methods. The novelty is limited.

---

> ### Author Response · Authors · 2021-11-18
> **Response to Reviewer Na7g (1/4)**
>
> Thank you for the detailed comments. We think a misunderstanding about the proposed positive sampling strategy and ProtoCL may have affected the reviewer's overall opinion on our paper. Before addressing your comments in detail, we briefly summarize the key points of our responses as follows.
>
> 1. The proposed method is not the combination of BYOL and PCL as the proposed ProtoCL is conceptually different from the PCL.
>
> 2. Formulating our method into an EM framework is not a key contribution of this paper; EM can provide more insights into our methods and make it easy to understand.
>
> 3. We politely point that both [1, 2] are fundamentally different from our positive sampling strategy as they both use a graph to select pseudo-positive examples that may not be truly positive.
>
> 4. The comparisons with [1] on various benchmark datasets are now included in the revision, further demonstrating the superiority of our method for deep clustering.
>
> Keeping these points in mind, we politely request a fair and complete re-evaluation of this paper as well as [1, 2].
>
> **Continued below...**

---

> > ### Author Response · Authors · 2021-11-18
> > **Response to Reviewer Na7g (2/4)**
> >
> > **...continued from Response (1/4) above**
> >
> > **`1. "Limited novelty as this paper simply combines two existing unsupervised learning methods, BYOL and PCL, and then applies it to the clustering task"`**
> >
> > We think a misunderstanding about the proposed ProtoCL may have affected the reviewer's opinion on the novelty of our paper. We must strongly emphasize that this paper does not simply combine BYOL and PCL. Our NCC is built upon BYOL with two novel and important techniques---positive sampling and ProtoCL loss---to further improve BYOL for deep clustering.
> >
> > 1. **Addressing the class collision issue for deep clustering with pure BYOL is NOT enough**. Although BYOL can avoid the class collision issue by removing negative examples, it mainly focuses on inducing transferable representations for the downstream tasks rather than grouping the data into different semantic classes in deep clustering. We note that BYOL still suffers from the representation collapse issues that need to be further addressed for the research community [5]. In addition, directly adding extra losses to BYOL for the deep clustering task would worsen the representation collapse since there are no negative examples to maintain a uniform representation space. In contrast, our NCC is built upon BYOL with the proposed positive sampling to improve within-cluster compactness and with the proposed ProtoCL to maximize inter-cluster distance.
> >
> > 2. **The proposed ProtoCL loss is conceptually different from ProtoNCE loss used in PCL**. The ProtoNCE used in PCL is a combination of instance-level contrastive loss and instance-to-prototypes contrastive loss; the former is to encourage instance uniformity while the latter one the within-cluster compactness. The ProtoNCE can be written as follows:
> >
> > $$\mathcal{L}\_{\mathrm{ProtoNCE}}=\mathcal{L}\_{\mathrm{NCE}}-\log \frac{\exp(\boldsymbol{\mu}\_{y\_i}^{\textrm T} \boldsymbol{v} / \tau)}{\sum\_{k=1}^{K} \exp( \boldsymbol{\mu}\_{k}^{\textrm T} \boldsymbol{v}/ \tau)},$$
> >
> > where $\boldsymbol{\mu}$ is the centroid obtained from k-means, $v$ the representation, $\tau$ the temperature, $y_i$ the clustering pseudo-labels. Clearly, PCL becomes a classification task under the supervision of pseudo-labels, which usually leads to improved cluster compactness. We would like to emphasize that PCL still needs another InfoNCE loss (the first term) to maintain the training stability. Without negative examples in BYOL, the prototypes would be also gradually collapsed during training, leading to collapse for BYOL-based deep clustering as well. Therefore, ProtoNCE still suffers the class collision issue, compromising the clustering performance. On the contrary, the proposed ProtoCL loss directly encourages the uniformity among clusters centers to maximize inter-cluster distance by optimizing
> > $$\mathcal{L}\_{\mathrm{ProtoCL}}=\frac{1}{K}\sum\nolimits\_{k=1}^{K} -\log \frac{\exp(\boldsymbol{\mu}\_k^{\textrm T} \boldsymbol{\mu}\_k^\prime / \tau)}{\exp(\boldsymbol{\mu}\_k^{\textrm T} \boldsymbol{\mu}^\prime\_k / \tau) + \sum\_{j=1, j\neq k}^{K}\exp(\boldsymbol{\mu}\_k^{\textrm T} \boldsymbol{\mu}\_j / \tau)}.$$
> > Since prototypes are mutually negative examples, ProtoCL loss has not introduced the class collision issue, leading to more uniform representations that improve BYOL for better clustering performance.
> >
> > 3. **Experimental results demonstrate that simply combining BYOL and PCL suffer from representation collapse**. We also provide experimental evidence in the following table. We visualize the Normalized Mutual Information and Standard Deviation (STD, $\times \sqrt{d}$ like SimSiam where $d$ is the feature dimension) of the $\ell_2$-normalized features. The high STDs indicate more uniform representations on the hypersphere. On one hand, compared to the MoCo-based PCL, applying ProtoNCE loss to BYOL drops the clustering performance since the representations collapsed as evidenced by a much lower STD. In other words, ProtoNCE still needs negative examples to avoid the representation collapse before applying it into BYOL. On the other hand, our proposed ProtoCL loss can well address this issue since it can encourage the uniformity of cluster centers, and further improve the clustering performance of BYOL by a significant margin.
> >
> > | Method   | Normalized Mutual Information | Standard Deviation |
> > | :------- | :---------------------------: | :----------------: |
> > | PCL+MoCo |         77.6$\pm$0.1          |       0.986        |
> > | PCL+BYOL |         74.4$\pm$2.3          |       0.375        |
> > | NCC      |         **85.1**$\pm$0.5          |       **0.990**        |
> >
> > In summary, we conclude that NCC is not the combination of BYOL and PCL. In addition to this, the rationale behind NCC is to improve the within-cluster compactness by the proposed positive sampling while maximizing the inter-class distance by the proposed ProtoCL, consequentially avoiding the representation collapse for BYOL.
> >
> > **Continued below...**

---

> > > ### Author Response · Authors · 2021-11-18
> > > **Response to Reviewer Na7g (3/4)**
> > >
> > > **...continued from Response (2/4) above**
> > >
> > > **`2. "The EM framework is also presented by the PCL"`**
> > >
> > > We would like to note that the EM framework is not the key contribution of our paper as EM framework is a common framework in clustering, not only presented in PCL but also in MiCE [3]. The reason why we formulate our method into an EM framework is to offer more insights about NCC and make it easy to understand.
> > >
> > > Although both in an EM framework, the M-step in PCL is significantly different from the one in our NCC. More specifically, the M-step in PCL is to optimize the ProtoNCE, which is an **instance-to-prototypes contrastive loss** to improve the within-cluster compactness while the M-step in our NCC is to optimize the proposed ProtoCL, which is a **prototypes-to-prototypes contrastive loss** to maximize the inter-cluster distance for better clustering performance. In addition, NCC also proposes a positive sampling strategy by sampling positive examples around each sample to improve within-cluster compactness. Finally, Table 3 demonstrates that NCC outperforms PCL by almost 10% AMI on the ImageNet-1k dataset.
> > >
> > >
> > >
> > > **`3. "References [1,2] thoroughly study the positive sampling strategy"`**
> > >
> > > We think a misunderstanding about the proposed positive sampling strategy may have affected the reviewer's opinion on the difference between references [1, 2] and our positive sampling strategy. We respectfully disagree that [1, 2] have studied the positive sampling strategy like ours, and note that [2] was made available in Arxiv after the ICLR submission deadline!
> > >
> > > Based on our understanding, GCC [1] and WCL [2] built a graph to label the neighbor samples as pseudo-positive examples. Then, they enforce the two data augmentations of one example to be close to its multiple pseudo-positive examples using a supervised contrastive loss. GCC [1] adopted a moving-averaged memory bank for the graph-based pseudo-labeling while WCL [2] built the graph within a mini-batch. References [1, 2] mainly focus on how to effectively select positive examples from mini-batch/memory bank to alleviate the class collision issue. However, they still suffer from the class collision issue as the selected pseudo-positive examples may not be truly positive. In addition to this, they still need negative examples for instance-wise contrastive learning.
> > >
> > > Here, we summarize the difference between our positive sampling strategy and theirs in the following four aspects.
> > >
> > > 1. References [1, 2] select the examples that exist in the dataset (mini-batch/memory bank) while ours samples augmented examples from the latent space that may not exist in the dataset.
> > >
> > > 2. References [1, 2] select neighbor examples in a graph as pseudo-positive examples that may not be truly positive while ours samples examples around the instance that can be assumed to be positive in the semantic space.
> > >
> > > 3. References [1, 2] still rely on instance-wise contrastive loss that could lead to class collision issue while ours can avoid class collision issue by using BYOL.
> > >
> > > 4. References [1, 2] require additional computational cost for graph construction while ours is rather cheap in sampling one augmented example.
> > >
> > >
> > >
> > > **`4. "Please also compare the results with these methods [1, 2]"`**
> > >
> > > Both methods [1, 2] handle the issue of inaccurate negative samples by building the graph to yield the pseudo-labels. However, they still need the contrastive loss and cannot completely avoid the class collision issue. The pseudo-positive examples selected in a graph may not be truly positive, still suffering class collision issue. Therefore, the pseudo-positive example selection method used in [1, 2] is fundamentally different from ours.
> > >
> > > Since reference [2] was made available in Arxiv after the ICLR submission deadline and did not release their source code, we reported the comparisons with reference [1] in the form of NMI/ACC/ARI in the following table. The results further demonstrate that our NCC outperforms reference [1] by a significant margin.
> > >
> > > | Method     |          CIFAR-10          |          CIFAR-20          |           STL-10           |        ImageNet-10         |       ImageNet-Dogs        |
> > > | ---------- | :------------------------: | :------------------------: | :------------------------: | :------------------------: | :------------------------: |
> > > | GCC [1]    |       76.4/85.6/72.8       |       47.2/47.2/30.5       |       68.4/78.8/63.1       |       84.2/90.1/82.2       |       49.0/52.6/36.2       |
> > > | NCC (ours) | **88.6**/**94.3**/**88.4** | **60.6**/**61.4**/**45.1** | **75.8**/**86.7**/**73.7** | **89.6**/**95.6**/**90.6** | **69.2**/**74.5**/**62.7** |
> > >
> > > **Continued below...**

---

> > > > ### Author Response · Authors · 2021-11-18
> > > > **Response to Reviewer Na7g (4/4)**
> > > >
> > > > **...continued from Response (3/4) above**
> > > >
> > > > **`5. "The comparison might be unfair. Some existing contrastive learning-based clustering methods, such as CC [4] and GCC [1], can also adopt the framework of BYOL and PCL to improve the results. The results might be unconvincing due to the limited novelty."`**
> > > >
> > > > We follow the widely-used settings in literature and believe that the comparisons are fair in terms of dataset splitting, backbone, training epochs, and so on.
> > > >
> > > > Existing contrastive learning-based clustering methods may not be applicable for non-contrastive learning frameworks such as BYOL as they mainly focus on introducing the cluster information into the contrastive loss. As suggested by the reviewer, CC performs cluster-level contrastive learning while GCC [1] uses the graph to select the pseudo-positive examples; they both need the instance-wise contrastive loss to avoid the representation collapse, inevitably leading to class collision issue.
> > > >
> > > > Although adapting CC and GCC [1] into the framework of BYOL and PCL yields new baselines for comparison, this is certainly beyond the scope of this paper. The compared baseline methods in this paper are recently published or available in Arxiv, representing the state-of-the-art performance for deep clustering. However, to convince you that existing contrastive learning-based methods may not be applicable for BYOL, we have tried our best to integrate CC [4] into BYOL in the following table. The results further demonstrate that directly applying CC into BYOL harms the clustering performance of vanilla BYOL. This is because CC performs a contrastive loss at an independent cluster head, which is not helpful for the representation learning of BYOL.
> > > >
> > > > | Method  |                      CIFAR-10                      |                      CIFAR-20                      |
> > > > | ------- | :------------------------------------------------: | :------------------------------------------------: |
> > > > | CC [4]  |         66.1$\pm$0.3/74.6$\pm$0.3/58.3±0.4         |       46.4$\pm$0.3/45.0$\pm$0.1/29.5$\pm$0.2       |
> > > > | BYOL    | **79.4**$\pm$1.7/**87.8**$\pm$1.7/**76.6**$\pm$2.8 | **55.5**$\pm$0.6/**53.9**$\pm$1.6/**37.6**$\pm$0.9 |
> > > > | BYOL+CC |       76.6$\pm$3.1/86.3$\pm$2.7/73.8$\pm$4.7       |       51.0$\pm$2.0/48.9$\pm$3.0/33.3$\pm$2.9       |
> > > >
> > > >
> > > >
> > > > ---
> > > >
> > > > [1] Graph Contrastive Clustering, ICCV 2021.
> > > >
> > > > [2] Weakly supervised contrastive learning, ICCV 2021.
> > > >
> > > > [3] MiCE: Mixture of Contrastive Experts for Unsupervised Image Clustering, ICLR 2021.
> > > >
> > > > [4] Contrastive clustering, AAAI 2021.
> > > >
> > > > [5] Understanding self-supervised learning dynamics without contrastive pairs, ICML 2021.

---

> > ### Comment · Area_Chair_EdkM · 2021-11-18
> > **On Response**
> >
> > Dear Authors,
> >
> > Thanks for your response. I understand you spent a lot of time and efforts to prepare the response. To make our internal discussion smooth, especially for some clarification questions and misunderstanding, here I have a little tip for your consideration. Post your opinion together with supportive evidence. For example, "the proposed ProtoCL is conceptually different from the PCL." In my eyes, this is an opinion. It would be better that you can provide the detailed difference between ProtoCL and PCL. By this means, the reviewer can help evaluate whether the evidence is enough to support your opinion.
> >
> > Note that the reviewers might increase or decrease their scores according to the response or new issues raised during the internal discussion.
> >
> > Kind Regards,
> >
> > AC

---

> > > ### Author Response · Authors · 2021-11-18
> > > **Response to AC**
> > >
> > > Dear AC,
> > >
> > > We really appreciate your tips on preparing responses. Due to our long response to Reviewer Na7g, we provide a brief summary of our response at the beginning. You can find the detailed responses below. As a side note, we will update our manuscript shortly with constructive suggestions from reviewers and additional experimental results.
> > >
> > > All the best,
> > >
> > > Authors

---

> > ### Comment · Reviewer_Na7g · 2021-11-20
> > **Response to Authors from Reviewer Na7g**
> >
> > Thanks very much for your detailed response and comparison. Sorry for the later reply due to the extension of CVPR DDL. After going through the papers as well as the response in detail, my concerns about the novelty still exist.
> >
> > (1) First, let me summarize the main contributions of this paper. Since the authors also agree in response that the EM framework is not a key contribution as it has been well studied in clustering, and the combination of BYOL is not a contribution, so the main contributions of this paper lie in two aspects, including the positive sampling strategy and ProtoCL.
> >
> > For the positive sampling, GCC (ICCV 2021)[1] has already adopted the nearest neighbors to construct positive samples for deep image clustering. This paper just use another simple strategy for augmentation. Therefore, the idea of positive sampling is not novel at all. Though the authors claim that positive samples constructed by the GCC may not be truely positive, but if we consider the background of BYOL without negative samples, the proposed strategy is just a simple and natural extension. Besides, the class collision issue is naturaly handled by BYOL instead of this paper.
> >
> > For the ProtoCL, it can be regarded as the cluster contrastive learning, which has been thoroughly investigated in CC and GCC. Please refer to the cluster-level contrastive head in Eq.(5) of CC[2] and assignment graph contrastive in Eq.(9) of GCC[1]. The only difference lies in the construction of cluster centers, which is just a minor revision. Though the authors compare with CC[2] in the response to Reviewer bq8Q, I still think this is a minor revision.
> >
> > Based on the above comparison, the idea of positive sampling and ProtoCL has already been well studied in deep image clustering, this paper just combines them with BYOL and makes minor revisions. Therefore, I think the novelty is quite limited for this top conference.
> >
> > (2) Second, though the reported results of the proposed method are very good on various datasets, I do believe that the combination of GCC and CC with BYOL or other architecture, e.g. MAE, can achieve competitive results under good parameter adjustment. With the fast development of unsupervised contrastive learning, eg. SimCLR, MoCo, BYOL, MAE, I think their simple extension to clustering with small revision does not make much contribution to this community.
> >
> > [1] Graph Contrastive Clustering. ICCV 2021
> > [2] Contrastive Clustering. AAAI 2021

---

> > > ### Author Response · Authors · 2021-11-20
> > > **Second Response to Reviewer Na7g (1/3)**
> > >
> > > We thank the reviewer again for appreciating the good performance on various datasets and requesting the clarification of the novelty of this work.
> > >
> > > **`1. The novelty is quite limited for this top conference.`**
> > >
> > > We summarize the novelty of this paper as follows.
> > >
> > > + We explored the non-contrastive representation learning for deep clustering, achieving a new state-of-the-art performance on various benchmark datasets. We hope this study will attract the community's attention to the important role of non-contrastive representation learning in deep clustering.
> > > + The proposed positive strategy sampling (augmented instance loss) is new in the following aspects.
> > >   + Assumption: one augmented view of one instance should be similar to **its neighboring examples of another view in the embedding space** in the context of clustering.
> > >   +  Loss: The augmented instance loss is to minimize the distance between one augmented view and **the Gaussian distribution of another view**; To simplify the computation, we sample one example from the Gaussian distribution with reparameterization trick.
> > >   + Purpose: It improves the within-cluster compactness.
> > > + The proposed prototypical contrastive loss is new in the following aspects.
> > >   + Assumption: The cluster centers should be pushed away in clustering.
> > >   + Loss: Contrastive loss over **prototypes** as for one cluster, remaining clusters are negative pairs. Positive pair comes from online network and target network in BYOL.
> > >   + Purpose: It improves the uniformity among clusters by maximizing the inter-cluster distance and improves the training stability of BYOL for deep clustering.
> > > + Although formulating our method into EM framework is not the key contribution of this work, it can facilitate the understanding of the training procedure.
> > >
> > > Keeping these points in mind, we believe our method is novel. Therefore, we have to politely request a fair and complete re-evaluation of this paper again.
> > >
> > >
> > >
> > > **`2. For the positive sampling, GCC (ICCV 2021)[1] has already adopted the nearest neighbors to construct positive samples for deep image clustering. This paper just uses another simple strategy for augmentation. Therefore, the idea of positive sampling is not novel at all.`**
> > >
> > > For effective discussion, we divide the class collision issue into the following two cases:
> > >
> > > + **Negative class collision issue**: negative examples are not truly negative, which is the case we discussed in the paper.
> > > + **Positive class collision issue**: positive examples are not truly positive, which is a new case raised in GCC [1]
> > >
> > > Then, let us use one sentence to summarize each method in terms of positive sampling/example selection as follows:
> > >
> > > + GCC leverages **a graph** to **select** data examples **from mini-batch/memory bank** as **pseudo-positive examples**, which **may not** be truly positive, **suffering** from **positive** class collision issue.
> > > + NCC leverages **a Gaussian distribution** to **sample** data examples **in the embedding space** as **positive examples**, which **can be assumed to** be truly positive, **avoiding** **both** class collision issues.
> > >
> > > Therefore, we think our positive sampling is significantly different from GCC. We also note that **simple** and **computational cheap** are merits of our method. We would like to clarify that our positive sampling is NOT augmentation; it performs sampling from a Gaussian distribution and computes an instance alignment loss.
> > >
> > > The ablation study in Table 4 demonstrates that the proposed positive sampling can stabilize the BYOL for better clustering performance. With positive sampling to improve the within-cluster compactness and ProtoCL to maximize the inter-cluster distance, the best results can be achieved.
> > >
> > > **Ablation study for positive sampling and ProtoCL (NMI/ACC/ARI)**
> > >
> > > | Method                 |                      CIFAR-10                      |                      CIFAR-20                      |
> > > | ---------------------- | :------------------------------------------------: | :------------------------------------------------: |
> > > | BYOL                   |       79.4$\pm$1.7/87.8$\pm$1.7/76.6$\pm$2.8       |       55.5$\pm$0.6/53.9$\pm$1.6/37.6$\pm$0.9       |
> > > | BYOL+Positive sampling |       79.4$\pm$0.9/87.9$\pm$0.5/76.4$\pm$1.1       |       57.0$\pm$0.0/55.0$\pm$0.6/39.8$\pm$1.1       |
> > > | BYOL+ProtoCL           |       83.4$\pm$1.2/90.3$\pm$0.9/81.1$\pm$1.7       |       56.6$\pm$0.4/55.1$\pm$0.5/40.7$\pm$1.0       |
> > > | NCC                    | **85.1**$\pm$0.5/**91.6**$\pm$0.4/**83.5**$\pm$0.7 | **58.2**$\pm$0.3/**57.8**$\pm$0.2/**42.3**$\pm$0.3 |
> > >
> > > **Continued below...**

---

> > > > ### Author Response · Authors · 2021-11-20
> > > > **Second Response to Reviewer Na7g (2/3)**
> > > >
> > > > **...continued from Second Response (1/3) above**
> > > >
> > > > **`3. Though the authors claim that positive samples constructed by the GCC may not be truly positive, but if we consider the background of BYOL without negative samples, the proposed strategy is just a simple and natural extension. Besides, the class collision issue is naturally handled by BYOL instead of this paper.`**
> > > >
> > > > We have to further clarify that GCC was based on the contrastive learning method in their original paper. As the reviewer insists on **that the proposed strategy is a simple and natural extension *if* one combines GCC and BYOL. We must strongly emphasize that although the negative class collision issue can be naturally handled by BYOL, the positive class collision caused by GCC still remains as the pseudo-positive samples selected from the dataset may not be truly positive.**
> > > >
> > > > We appreciate the reviewer's critical comment, which reminds us of the two different kinds of class collision issues and makes the difference clear!
> > > >
> > > > **`4. For the ProtoCL, it can be regarded as the cluster contrastive learning, which has been thoroughly investigated in CC and GCC. Please refer to the cluster-level contrastive head in Eq.(5) of CC[2] and assignment graph contrastive in Eq.(9) of GCC[1]. The only difference lies in the construction of cluster centers, which is just a minor revision.`**
> > > >
> > > > Again, let us use one sentence to summarize each method in terms of class-level contrastive loss as follows
> > > >
> > > > + Eq. (5) in CC [2] performs contrastive learning over **cluster probabilities** of **two data augmentations**.
> > > >
> > > > + Eq. (9) in GCC performs contrastive learning over **assignment probabilities** of **one data augmentation and one pseudo-positive example**.
> > > > + ProtoCL in NCC performs contrastive learning over **prototypes** of **two data augmentations**.
> > > >
> > > > We think that the cluster/assignment probabilities would lose the semantic information of the learned representations, which is not helpful for representation learning. In contrast, our ProtoCL is **the first** to implement contrastive loss on **the representation of cluster centers** to maximize the inter-cluster distance, which can sense the semantic information of the latent space. In addition, Eq. (9) in GCC would still suffer from positive class collision issue as the selected pseudo-positive example from the graph may not be truly positive.
> > > >
> > > > The difference can also be observed in Table 4; for your conveninece, we put part of it below. BYOL + ProtoCL is significantly better than BYOL + CC.
> > > >
> > > > **Comparisons between CC and ProtoCL (NMI/ACC/ARI)**
> > > >
> > > > | Method       |                      CIFAR-10                      |                      CIFAR-20                      |
> > > > | ------------ | :------------------------------------------------: | :------------------------------------------------: |
> > > > | BYOL         |       79.4$\pm$1.7/87.8$\pm$1.7/76.6$\pm$2.8       |       55.5$\pm$0.6/53.9$\pm$1.6/37.6$\pm$0.9       |
> > > > | BYOL+CC      |       76.6$\pm$3.1/86.3$\pm$2.7/73.8$\pm$4.7       |       51.0$\pm$2.0/48.9$\pm$3.0/33.3$\pm$2.9       |
> > > > | BYOL+ProtoCL | **83.4**$\pm$1.2/**90.3**$\pm$0.9/**81.1**$\pm$1.7 | **56.6**$\pm$0.4/**55.1**$\pm$0.5/**40.7**$\pm$1.0 |
> > > >
> > > > **`5. I do believe that the combination of GCC and CC with BYOL or other architecture, e.g. MAE, can achieve competitive results under good parameter adjustment.`**
> > > >
> > > > As we stated in the previous [response (4/4)](https://openreview.net/forum?id=JZrETJlgyq&noteId=YnUT4PqUHb6) on Nov 19, adapting GCC [1] and CC [2] with BYOL or other architecture, *e.g.* MAE [4], is beyond the scope of this paper.
> > > >
> > > > We would like to point out that CC [2] and GCC [1] were made available in Arxiv on **September 21, 2020**, and **April 3, 2021**, respectively. However, BYOL was made available in Arxiv on **June 3, 2020, earlier than both papers**.  There may be a reason why CC and GCC are based on contrastive learning rather than BYOL. We also provided the results of combining CC with BYOL, which does not achieve competitive results as the reviewer expects, even worse than BYOL itself.
> > > >
> > > > We would like to further point out that MAE [4] was made available in Arxiv on **November 11, 2021**. We believe that there are no MAE-based clustering methods available for comparison so far.
> > > >
> > > > We have also conducted many experiments on different hyper-parameters, which demonstrate that NCC is robust to the choice of hyper-parameters.
> > > >
> > > > **Continued below...**

---

> > > > > ### Author Response · Authors · 2021-11-20
> > > > > **Second Response to Reviewer Na7g (3/3)**
> > > > >
> > > > > **...continued from Second Response (2/3) above**
> > > > >
> > > > > **`6. With the fast development of unsupervised contrastive learning, eg. SimCLR, MoCo, BYOL, MAE, I think their simple extension to clustering with small revision does not make much contribution to this community.`**
> > > > >
> > > > > We agree with you that unsupervised contrastive learning is being developed very fast. However, we would like to respectfully disagree with you that **BYOL and MAE [4] belong to the category of unsupervised contrastive learning** as they are non-contrastive learning methods.
> > > > >
> > > > > Our results suggest that non-constrative learning outperforms contrastive learning for deep clustering, which could attract the community's attention to the non-contrastive learning for deep clustering. We think this is an important contribution to the community.
> > > > >
> > > > > In the future, we will investigate the MAE-based clustering algorithm and see if this new architecture can bring more exciting results to the deep clustering community.
> > > > >
> > > > > **Comparisons between contrastive and non-contrastive learning methods for deep clustering (NMI/ACC/ARI)**
> > > > >
> > > > > | Method  |                      CIFAR-10                      |                      CIFAR-20                      |
> > > > > | ------- | :------------------------------------------------: | :------------------------------------------------: |
> > > > > | MoCo v2 |       76.9$\pm$0.2/84.9$\pm$0.3/72.4$\pm$0.5       |       49.2$\pm$0.1/48.0$\pm$0.2/32.1$\pm$0.0       |
> > > > > | BYOL    | **79.4**$\pm$1.7/**87.8**$\pm$1.7/**76.6**$\pm$2.8 | **55.5**$\pm$0.6/**53.9**$\pm$1.6/**37.6**$\pm$0.9 |
> > > > >
> > > > >
> > > > >
> > > > > To conclude, we appreciate the reviewer's critical comments, which helped improve the quality of this paper. We are looking forward to further feedbacks. Thanks!
> > > > >
> > > > > ---
> > > > >
> > > > > [1] Zhong, Huasong, Jianlong Wu, Chong Chen, Jianqiang Huang, Minghua Deng, Liqiang Nie, Zhouchen Lin, and Xian-Sheng Hua. "Graph Contrastive Clustering." ICCV 2021.
> > > > >
> > > > > [2] Li, Yunfan, Peng Hu, Zitao Liu, Dezhong Peng, Joey Tianyi Zhou, and Xi Peng. "Contrastive clustering." AAAI 2021.
> > > > >
> > > > > [3] Zheng, Mingkai, Fei Wang, Shan You, Chen Qian, Changshui Zhang, Xiaogang Wang, and Chang Xu. "Weakly supervised contrastive learning." ICCV 2021.
> > > > >
> > > > > [4] He, Kaiming, Xinlei Chen, Saining Xie, Yanghao Li, Piotr Dollár, and Ross Girshick. "Masked Autoencoders Are Scalable Vision Learners." arXiv preprint arXiv:2111.06377 (2021).

---

### Official Review · Reviewer_bq8Q · 2021-11-02

**Correctness:** 4
**Technical Novelty And Significance:** 3
**Empirical Novelty And Significance:** 3
**Recommendation:** 6
**Confidence:** 4

**Main Review:**

1. strengths

-the authors provide a sufficient literature review.
-the paper is well organized.
-detailed appendix, including ELBO, convergence analysis, experimental setup, etc.
-abundant experimental results.
-large performance improvement.

2. weaknesses

-the non-contrastive representation motivation maybe overstated. L_{aug-ins} is a non-contrastive loss, but L_{pcl} is a typical contrastive loss. From Tab.4, L_{pcl} may be more important than L_{aug-ins}.
-the proposed L_{pcl} is not novel enough. This loss is actually class-level contrastive loss, which is proposed in the work (Contrastive clustering, AAAI 2021)
-the EM-based framework is not novel enough. ProtoNCE firstly adopt this framework in representation clustering related area.
-the reviewer would like to see more clustering results in some challenging datasets, such as Tiny-ImageNet, ImageNet-50/100/200 subsets (in SCAN, ECCV 2020).


**Summary Of The Paper:**

This paper proposes a novel deep clustering method with non-contrastive representation motivation. The authors provide detailed experimental results to show the superior performance of the proposed method.

**Summary Of The Review:**

Although the reviewer provides some negative feedback in Main Review, the reviewer still likes this paper. Therefore, I will give a positive rating.

---

> ### Author Response · Authors · 2021-11-18
> **Response to Reviewer bq8Q (1/2)**
>
> Thank you for the encouraging and valuable feedback.
>
> **`1. "The non-contrastive representation motivation maybe overstated. L_{aug-ins} is a non-contrastive loss, but L_{pcl} is a typical contrastive loss. From Tab.4, L_{pcl} may be more important than L_{aug-ins}."`**
>
> We agree with you that $\mathcal{L}\_{\mathrm{aug-ins}}$ is a non-contrastive loss and $\mathcal{L}\_{\mathrm{pcl}}$ is a contrastive loss. We would like to clarify that our non-contrastive representation motivation lies in the representation learning part directly related to $\mathcal{L}\_{\mathrm{aug-ins}}$. Although $\mathcal{L}\_{\mathrm{pcl}}$ is a contrastive loss,  it operates on cluster centers instead of instances, which contributes to the goodness of clustering. Therefore, we would like to keep non-contrastive representation motivation; however, we are open to revise it if there are better ideas to improve it.
>
> When we only use one loss, $\mathcal{L}\_{\mathrm{pcl}}$ may be more important than $\mathcal{L}\_{\mathrm{aug-ins}}$ as shown in Table 4. This is because that $\mathcal{L}\_{\mathrm{pcl}}$ can directly maximize inter-cluster distance for better clustering performance. However, when combining them together, the performance can be better.
>
> **`2. "The proposed L_{pcl} is not novel enough. This loss is actually a class-level contrastive loss, which is proposed in the work [1]"`**
>
> Although both $\mathcal{L}\_{\mathrm{pcl}}$ and CC [1] are class-level contrastive loss, which perform contrastive learning at the cluster level, they have the following difference.
>
> 1. **The class-level contrastive loss in CC [1] implements the contrastive loss on the cluster probabilities while ours on the representation of cluster centers**. Implementing contrastive loss on the cluster probability in [1] would lose the semantic information of the learned representations, which is not helpful for representation learning. Specifically, given the $\boldsymbol{x} \in \mathcal{B}$, CC [1] obtains the cluster assignments $\boldsymbol{P}\_k=[p(k \vert \boldsymbol{x}\_{1}), \cdots, p(k \vert \boldsymbol{x}\_{N})]$ from one view and $\boldsymbol{P}\_{k}^{'}$ from another view, and then contrasts $\boldsymbol{P}\_k$ and $\boldsymbol{P}\_{k}^{'}$ at the cluster level using the InfoNCE loss. In contrast, $\mathcal{L}\_{\mathrm{pcl}}$ implements the contrastive loss on the representation of the cluster centers within a mini-batch using the pseudo-labels from k-means clustering. As a result, $\mathcal{L}_{\mathrm{pcl}}$ is able to sense the semantic information of the latent space and make the representations of clusters more discriminative and suitable for the clustering task.
>
> |                              | Contrast objects                                             |
> | ---------------------------- | ------------------------------------------------------------ |
> | CC [1] | cluster probability $$\boldsymbol{P}\_k=[p(k \vert \boldsymbol{x}\_{1}), \cdots, p(k \vert \boldsymbol{x}\_{N})]$$ |
> | $\mathcal{L}\_{\mathrm{pcl}}$ (ours) | the representations  of cluster centers  $$\boldsymbol{\mu}\_{k} =\frac{\sum\_{\boldsymbol{x} \in \mathcal{B} }  p(k\| \boldsymbol{x}) f(\boldsymbol{x}) }{\vert\vert\sum\_{\boldsymbol{x} \in \mathcal{B} }  p(k\vert\vert \boldsymbol{x}) f(\boldsymbol{x})\vert\vert\_{2}}$$ |
>
> 2. **The class-level contrastive loss in CC [1] does not encourage cluster uniformity while our ProtoCL does.** CC [1] still needs the instance-wise contrastive loss to encourage the instance uniformity, which inevitably introduces the class collision issue. In the revised manuscript, we have also conducted experiments that integrate CC into BYOL. The performance of BYOL drops and the training becomes unstable. Under the same conditions, NCC achieves significant improvements over CC [1].
>
> **Continued below...**

---

> > ### Author Response · Authors · 2021-11-18
> > **Response to Reviewer bq8Q (2/2)**
> >
> > **...continued from Response (1/2) above**
> >
> > **`"3. The EM-based framework is not novel enough. ProtoNCE firstly adopt this framework in representation clustering related area"`**
> >
> > We would like to note that the EM framework is not the key contribution of our paper as EM framework is a common framework in clustering, not only presented in PCL but also in MiCE [2]. The reason why we formulate our method into an EM framework is to offer more insights about NCC and make it easy to understand.
> >
> > Although both in an EM framework, the M-step in PCL is significantly different from the one in our NCC. More specifically, the M-step in PCL is to optimize the ProtoNCE, which is an **instance-to-prototypes contrastive loss** to improve the within-cluster compactness while the M-step in our NCC is to optimize the proposed ProtoCL, which is a **prototypes-to-prototypes contrastive loss** to maximize the inter-cluster distance for better clustering performance. In addition, NCC also proposes a positive sampling strategy by sampling positive examples around each sample to improve within-cluster compactness. Finally, Table 3 demonstrates that NCC outperforms PCL by almost 10% AMI on the ImageNet-1k dataset.
> >
> > **`4. "The reviewer would like to see more clustering results in some challenging datasets, such as Tiny-ImageNet,
> > ImageNet-50/100/200 subsets [3]" `**
> >
> > Thank you for your constructive comment. We would like to note that ImageNet-1K is deemed to be the most challenging dataset for clustering as it contains 1K classes. On this dataset, our method achieves the state-of-the-art performance over baseline methods by a large margin.
> >
> > Per your suggestion, we further added the results of NCC on these three datasets. Under fair comparisons, NCC consistently achieves better clustering performance by a significant margin.
> >
> > **Comparisons on Tiny-ImageNet (NMI/ACC/ARI)**
> >
> > | Method     | Tiny-ImageNet              |
> > | ---------- | -------------------------- |
> > | CC [1]     | 34.0/14.0/7.1              |
> > | GCC [4]    | 34.7/13.8/7.5              |
> > | NCC (Ours) | **40.5**/**25.6**/**14.3** |
> >
> > **Comparisons on ImageNet subsets (NMI/ARI) over SCAN**
> >
> > | Method     | 50 Classes        | 100 Classes       | 200 Classes       |
> > | ---------- | ----------------- | ----------------- | ----------------- |
> > | SCAN [3]   | 82.2/66.1         | 80.8/57.6         | 77.2/47.0         |
> > | NCC (Ours) | **82.8**/**69.1** | **83.5**/**63.5** | **80.6**/**53.8** |
> >
> > We note that we train NCC for 300 epochs while SCAN has used the pre-trained model of MoCo trained on the full ImageNet for 800 epochs.
> >
> > [1] Contrastive clustering, AAAI 2021.
> >
> > [2] MiCE: Mixture of Contrastive Experts for Unsupervised Image Clustering. ICLR 2021.
> >
> > [3] SCAN, ECCV 2020.
> >
> > [4] Graph Contrastive Clustering, ICCV 2021.

---

### Author Response · Authors · 2021-11-19
**Revision Summary**

We thank the reviewers for their feedback, especially *bq8Q*, *dith*, and *nh13* for appreciating the significance of our work and providing constructive and encouraging suggestions to improve the quality of this work. In addition to the detailed responses to all comments from reviewers, we have made necessary changes to the initial manuscript.

Here, we summarize the changes in the revised manuscript, which have been highlighted in blue.

+ We improved the presentation of this paper in a more top-down way. (in response to *Reviewer nh13*)
+ In Table 2, we reported the results of PCL. (in response to *Reviewer nh13*)
+ In Table 2, we reported the results of GCC. (in response to *Reviewer Na7g*)
+ In Table 4, we reported the results of CC and PCL for ablation study, which were then integrated into BYOL.  (in response to *Reviewers Na7g* and *nh13*)
+ In Appendix A and Related Work, we provided the detailed difference between ours and existing clustering methods including CC, GCC, WCL, and PCL. (in response to *Reviewers bq8Q*, *Na7g*, and *nh13*)
+ In Appendix C, we provided the implementation details of CC and PCL. (in response to *Reviewers Na7g* and *nh13*)
+ In Appendix Figure A1, we analyzed that why PCL may not be applicable to BYOL.  (in response to *Reviewers Na7g* and *nh13*)
+ In Appendix Figure A6, we further analyzed the over-clustering behaviors on the two datasets including NCC and BYOL. (in response to *Reviewer dith*)
+ In Appendix Figure A7, we visualized the training and cluster statistics of NCC and BYOL, and discussed that the class collision issue arose from k-means clustering may be alleviated in NCC. (in response to *Reviewer dith*)
+ In Appendix Tables A2 and A3, we conducted extra experiments on Tiny-ImageNet and three ImageNet subsets used in SCAN. (in response to *Reviewer bq8Q*)

------
updated on Nov. 23

+ In Table 4, we reported the results of integrating the proposed ProtoCL into CC for the fair comparisons of self-supervised learning frameworks. (in response to *Reviewers bq8Q* and *Na7g*)

---

### Author Response · Authors · 2021-11-23
**Additional comparison between class-level contrastive loss in CC and the proposed ProtoCL**

In response to *Reviewer bq8Q* and *Reviewer Na7g*, we have provided the detailed difference between the class-level contrastive loss in CC [1] and the proposed ProtoCL, and the experimental comparison between them based on the same non-contrastive learning (BYOL) framework. Here, we further provide the experimental comparison between them based on the same contrastive learning framework as follows.

Specifically, for fair comparisons of the self-supervised learning framework, we integrate our ProtoCL into CC by replacing the cluster head with our ProtoCL on the representations while keeping other official hyper-parameters unchanged. The results are presented in the following table.

Although class collision issue remains, the significant improvements over CC on both datasets support our claim that the class-level contrastive loss over representations of cluster centers is better than the one over cluster probabilities. In addition, the proposed ProtoCL can be generalized to other self-supervised learning frameworks for better clustering performance.

To conclude, no matter which self-supervised learning framework the deep clustering is based on, the results show that the proposed ProtoCL can achieve better performance than the class-level contrastive loss in CC [1].



**Comparisons between CC and CC+ProtoCL (NMI/ACC/ARI)**

| Method     | CIFAR-10                                           | CIFAR-20                                           |
| ---------- | -------------------------------------------------- | -------------------------------------------------- |
| CC         | 66.1$\pm$0.3/74.6$\pm$0.3/58.3$\pm$0.4             | 46.4$\pm$0.3/45.0$\pm$0.1/29.5$\pm$0.2             |
| CC+ProtoCL | **74.3**$\pm$0.4/**83.4**$\pm$0.5/**69.6**$\pm$1.0 | **48.3**$\pm$0.2/**49.1**$\pm$0.2/**32.2**$\pm$0.4 |

[1] Li, Yunfan, Peng Hu, Zitao Liu, Dezhong Peng, Joey Tianyi Zhou, and Xi Peng. "Contrastive clustering." AAAI 2021.

---

### Decision · Program_Chairs · 2022-01-20

**Decision:**

Reject

**Comment:**

This paper received the initial scores with large variance. During the intensive discussion (Number of Forum replies is up to 60), the opinions reached the consensus. I have read all the materials of this paper including manuscript, appendix, comments and response. Based on collected information from all reviewers and my personal judgement, I can make the recommendation on this paper, *rejection*. Here are the comments that I summarized, which include my opinion and evidence.

**Research Problem and Motivation**

(1) It seems that the authors aimed to address the question that “are negative examples necessary for deep clustering?” This research problem has been proposed and addressed in BYOL and SimSiam (If I remembered correctly, some reviewer pointed this out). What the authors actually did is to add two more components, positive sampling strategy and prototypical contrastive loss, on the top of BYOL. In my eyes, it is like putting two patches on BYOL, where one of them does not work (I will explain later).

(2) Moreover, the authors failed to clearly illustrate the drawback of BYOL. In the last sentence of the third paragraph in the Introduction part, the authors mentioned that “BYOL only optimize the alignment term, leading to unstable training and suffering from the representation collapse.” This sentence is too general, which lacks strong motivation.

Therefore, the research problem addressed here is an incremental problem over BYOL, rather than brings new insights into the contrastive learning community.

**Philosophy**

Without a clear motivation, it is difficult to catch the philosophy of this paper, i.e., how their proposed components tackle BYOL’s drawbacks. Moreover, the relationship between two components is also unclear.

**Novelty**

I believe Reviewer Na7g has a thorough analysis of the novelty of this paper. I will not go into details here. The difference does not mean novelty.

**Technique**

Positive sampling strategy does not work. If we take a closer look at Table 4, the rows of BYOL and NCC with PS, there is no significant performance gain. The p-values of t-test on ACC results on CIFAR-10 and CIFAR-20 are 0.92 and 0.32, respectively. Actually, the prototypical contrastive loss is the key element to boost performance over BYOL.

**Misleading Title**

Based on the above point, the title is misleading. Although no negative data sample pairs are used in the training, the contractiveness on the cluster-level should also belong to the scope of contrastive learning.

**Experiments**

(1) In the Introduction part, the authors mentioned that SimSiam is in the same non-contrastive category with this paper. However, this paper is not included in the comparison.

(2) The competitive methods in Table 2 and 3 are not consistent. The authors even did not report the performance of BYOL on ImageNet-1K.

(3) Positive sampling strategy does not work. See the above Technique point.

(4) The authors only reported the running time on CIFAR-10 and CIFAR-20.

Therefore, the experimental results are not very convincing and solid to me.

**Presentation**

I believe the presentation also needs many efforts to smooth the logic. For example, “Even though Grill et al. (2020); Richemond et al. (2020) have proposed to use some tricks such as SyncBN (Ioffe & Szegedy, 2015) and weight normalization (Qiao et al., 2019) to alleviate this issue, the additional computation cost is significant.” Actually, Grill et al. (2020) is BYOL, where the authors added their components on. The computational cost of the proposed method should be heavier than BYOL.

Based on the above points, this paper suffers from several severe issues, which makes it not self-standing.